# Targeting ALK averts ribonuclease 1-induced immunosuppression and enhances antitumor immunity in hepatocellular carcinoma

Chunxiao Liu [1,2,7] ✉, Chenhao Zhou[2,3,7], Weiya Xia[2,4], Yifan Zhou[5,6], Yufan Qiu[2], Jialei Weng[3], Qiang Zhou[3], Wanyong Chen[3], Ying-Nai Wang [2], Heng-Huan Lee [2], Shao-Chun Wang [4], Ming Kuang [1], Dihua Yu [2], Ning Ren [3] ✉ & Mien-Chie Hung [2,4] ✉

Tumor-secreted factors contribute to the development of a microenvironment that facilitates the escape of cancer cells from immunotherapy. In this study, we conduct a retrospective comparison of the proteins secreted by hepatocellular carcinoma (HCC) cells in responders and non-responders among a cohort of ten patients who received Nivolumab (anti-PD-1 antibody). Our findings indicate that non-responders have a high abundance of secreted RNase1, which is associated with a poor prognosis in various cancer types. Furthermore, mice implanted with HCC cells that overexpress RNase1 exhibit immunosuppressive tumor microenvironments and diminished response to anti-PD-1 therapy. RNase1 induces the polarization of macrophages towards a tumor growth-promoting phenotype through activation of the anaplastic lymphoma kinase (ALK) signaling pathway. Targeting the RNase1/ALK axis reprograms the macrophage polarization, with increased CD8⁺ T- and Th1- cell recruitment. Moreover, simultaneous targeting of the checkpoint protein PD-1 unleashes cytotoxic CD8⁺ T-cell responses. Treatment utilizing both an ALK inhibitor and an anti-PD-1 antibody exhibits enhanced tumor regression and facilitates long-term immunity. Our study elucidates the role of RNase1 in mediating tumor resistance to immunotherapy and reveals an RNase1-mediated immunosuppressive tumor microenvironment, highlighting the potential of targeting RNase1 as a promising strategy for cancer immunotherapy in HCC.

Hepatocellular carcinoma (HCC) is one of the most common solid malignancies worldwide. Multiple strategies have been approved for first or second-line treatment for advanced HCC, such as sorafenib, cabozantinib, lenvatinib, atezolizumab plus bevacizumab, tremelimumab plus durvalumab, and others[1–3]. However, the therapeutic effects of those agents are far from satisfactory. Nivolumab (an anti-PD-1 mAb) was approved for advanced HCC treatment based on the results of CheckMate-040[4]. However, only a minority of patients appear to benefit from ICI monotherapy[5]. Several gene signatures which were related to cytolytic activity (GZMA and PRF1)[6], inflammatory activity

(CD274, CD8A, LAG3, and STAT1)[7], IFN-γ/IRF-1 pathway[8], antigen presentation (HLA-related genes)[9], T-cell exhaustion (TIM3, CTLA4, LAG3, CD39 and TIGIT)[10] and activation of natural killer cells[11], are demonstrated to be critical features of HCCs responding to anti-PD-1. However, the benefit conveyed by these biomarkers in clinic is limited. Although several novel combinations are in development, including anti-PD-1/PD-L1 antibodies in combination with anti-CTLA4 and/or anti-VEGF antibodies or multikinase inhibitors[12], the lack of predictive biomarkers to guide the use of those combinational strategies dilute the efficacy. Thus, it is of high unmet need to provide biomarkers to identify specific population of patients for right combination therapy.

The cross-talk between cancer and the immune system plays a critical role in cancer progression and response to immunotherapies. For HCC in particular, although it is immunogenic cancer that expresses various tumor-associated antigens and is characterized by an extensively abundant infiltrating immune cell microenvironment, immunotherapy approaches such as immune checkpoint inhibition alone have not demonstrated meaningful effectiveness against HCC, and the availability of predictive biomarkers is limited[13–15]. The tumor microenvironment (TME) acts as a major barrier for the recruitment and activation of effector lymphocytes[16]. The composition and physical properties of the TME are regulated by secreted factors during tumor development, and the proteins secreted by cancer cells play pivotal roles in this regulation[17–19]. Thus, strategies targeting cancer cell-secreted factors can be exploited to not only inhibit cancer cell proliferation and metastasis but also maximize the benefit of immunomodulatory therapies. However, the complexity of secreted proteins makes obtaining a clear picture of how tumor cells contribute to defense against immunotherapy difficult.

RNases are secreted in the circulatory system and are generally recognized as part of the host defense against pathogens[20]. Despite the discovery of high expression of RNases in multiple types of cancer cells[21–23], which acts as a ligand to mediate physiologically and pathologically relevant cellular events, including cancer cell growth[22], cancer cell stemness[23] and regenerative capabilities[24], little is known about the regulation and function of RNases in TMEs. Therefore, we performed whole-transcriptome analysis with total RNA-sequencing of human HCC samples from nivolumab (anti-PD-1) responders and non-responders.

In this work, we identify higher expression of RNase1 in non-responders than in responders, and that the RNase1 level is associated with poor survival across several types of cancer. Moreover, RNase1 reprograms macrophage polarization by regulating anaplastic lymphoma kinase (ALK)-mediated signaling and transcriptional events. Our study provides an understanding of the cross-talk between tumor and immune cells in the TME, and identifies RNase1 as a potential plasma biomarker to overcome tumor-associated macrophage (TAM)-mediated immunosuppression in HCC patients.

## Results
### RNase1 predicts immunotherapy resistance of HCC
To identify critical tumor-secreting factors that may contribute to immunotherapy efficacy against HCC, we subjected tumor samples from 10 patients with pathologically confirmed HCC to whole-transcriptome sequencing, and categorized the patients into two groups according to clinical response to nivolumab (Fig. 1a, Supplementary Fig. 1a, Supplementary Table 1). Gene Ontology (GO) analysis showed that upregulated genes in non-responders primarily played a role in immune response (Fig. 1b), whereas upregulated genes in responders were mainly involved in redox reactions and metabolic processes (Supplementary Fig. 1b). We further performed Gene set variation analysis (GSVA) analysis to screen immune-related biological differences between responders and non-responders, and we found that the immunologic pathway related to monocyte to M2-like macrophage differentiation was enriched in non-responders. In addition, CD4[+] T-cell, myeloid cell, B cell-related immunologic pathways are enriched in responders (Supplementary Fig. 1c). Transcriptome analysis revealed a set of significantly differentially expressed secreted genes, including AFP, MMP9, MMP-7, and S100A9, which are known regulators of cancer progression and therapeutic targets in cancer patients[25–27] (Fig. 1c). Among the 34 most significantly enriched secreted proteins ($\log_2$ fold change ≥ 2, $P < 0.001$) (Fig. 1d, Supplementary Table 2), we found that RNase1, which belongs to the human Ribonuclease A superfamily, was one of the 3 genes (RNase1, MMP9 and NPW) that was upregulated in all of 5 non-responders (5/5, 100%) than in responders (Supplementary Fig. 1d). Among the three genes, RNase1 and MMP9 are correlated with poor prognosis in HCC TCGA database (Supplementary Fig. 1e), further literature survey indicated that MMP9 is known to modulate immunosuppression in tumor[28,29]. Considering that RNase1 and other RNases were significantly associated with overall survival in large patient cohorts in different cancer types[21–23], RNase1 was the top candidate for further investigation due to its upregulation in non-responders receiving Nivolumab and its unknown function in tumor immunity.

To validate the clinical relevance of RNase1 in patients with HCC, we performed a tissue microarray-based immunohistochemical study of RNase1 expression in 174 HCC samples (Supplementary Table 3). We observed that patients with high RNase1 expression had markedly shorter overall survival (OS) and time to recurrence than did patients with low expression (Fig. 1e). Moreover, cox proportional-hazards regression analysis demonstrated that RNase1 status was an independent predictor of OS and RFS in HCC cohort (Supplementary Table 4). In addition to HCC, it would be interesting to see whether RNase1 is a potential biomarker to predict patient outcomes in other malignancies, especially in the cancer types including breast cancer[23], esophageal squamous cell carcinoma, lung squamous cell carcinoma, stomach adenocarcinoma and thymoma which have significantly correlation between RNase1 and poor prognosis according to KM plotter analyses results (Supplementary Fig. 1f). Other risk factors might play critical roles in the development of those certain types of cancer which has no significant correlation between RNase1 and poor outcome (Supplementary Fig. 1g).

To evaluate the impact of tumor-cell-secreted RNase1 on the TME, we measured the levels of RNase1 in HCC cell lines. We found that RNase1 was expressed in the tumor-cell-conditioned medium derived from six out of nine HCC cell lines (Hepa1-6, HepG2, HA22T, HA59T, Tong, and Mahlavu) (Fig. 1f). Notably, we detected considerably higher RNase1 plasma levels in HCC patients than in normal individuals (~0.4 µg/ml)[23,30] (Fig. 1g) and a strong positive correlation of RNase1 expression in HCC patients' tumor sections with that in paired plasma samples ($R = 0.60$; Fig. 1h), suggesting that tumor cells are major sources of increased RNase1 expression in HCC patients. In addition, we found patients with higher plasma RNase1 level (≥Median) had significantly shorter OS duration ($P = 0.017$) and time to recurrence ($P = 0.014$; Fig. 1i), suggesting that plasma RNase1 is a good potential biomarker for clinical selection. To further demonstrate the potential prognostic value of RNase1 in patients with HCC who receive immunotherapy, we examined the expression of RNase1 and the infiltration of CD8[+] T cells in another independent HCC cohort (Supplementary Table 5). We found that the patients who had responses to nivolumab exhibited lower RNase1 expression and greater CD8[+] T-cell infiltration than did non-responders (Fig. 1j). Furthermore, the negatively correlation between CD8[+] T-cell infiltration and RNase1 expression (Supplementary Fig. 1h) prompt us to ask whether RNase1 might associate with immunosuppressive tumor microenvironment in HCC.

### RNase1 overexpression correlates with an immunosuppressive TME
To validate the role of RNase1 in anti-PD-1 therapy efficacy in vivo, we selected HCA-1 line which does not express RNase1 as shown in Fig. 1f and generated RNase1-expressing HCA-1 stable cell lines (Supplementary

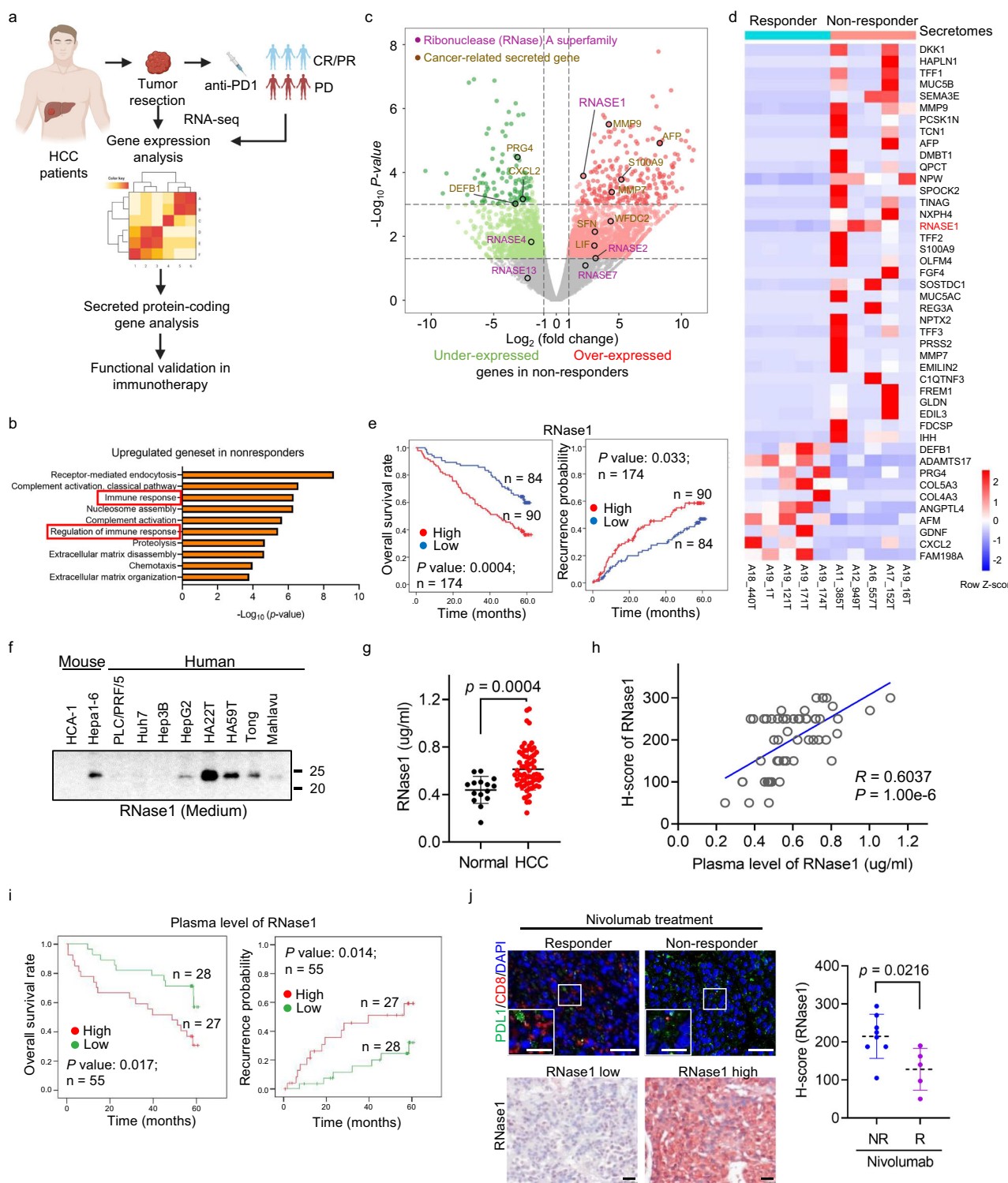

Fig. 2a) and inoculated the RNase1-expressing HCA-1 (HCA-1/R1) or HCA-1-vector (HCA-1/Vec) cells into immunocompetent mice, which was followed by treatment with a mouse anti-PD-1 antibody (αPD-1), that was previously used in preclinical studies (Supplementary Fig. 2b)[31]. Compared with the mice bearing HCA-1/Vec tumors, the mean plasma level of RNase1 was substantially higher in mice bearing HCA-1/R1 tumors (0.52 μg/ml) (Fig. 2a); this level was similar to the level found in the HCC patients described above (0.61 μg/ml) (Fig. 1g). Although the tumor size has significant difference between IgG and anti-PD-1 treatment in mice with HCA-1/R1, the efficacy of αPD-1 was much greater in mice with HCA-1/Vec (Supplementary Fig. 2c). In addition, we compared the efficacy of

αPD-1 in the indicated groups by monitoring mice overall survival rate and found that αPD-1 treatment did not prolong overall survival rate in mice with HCA-1/R1 (Supplementary Fig. 2d), supporting the resistant role of RNase1. Furthermore, we examined the therapeutic responses to anti-PD-1 using an HCA-1/Vec or HCA-1/R1 orthotopic model (Fig. 2b). Consistent with the subcutaneous HCC mouse model, after 3 weeks of anti-PD-1 treatment, compared with the mice bearing HCA-1/Vec tumors, mice bearing HCA-1/R1 tumors exhibited a significantly enhanced tumor size and a poor therapeutic response (Fig. 2c–e), indicating that over-expression of RNase1 promoted HCC cell highly resistance to anti-PD-1 treatment.

**Fig. 1 | RNase1 predicts poor prognosis for HCC and poor clinical response of it to anti-PD-1 therapy. a** Experimental strategy. Figure was created with BioRender.com. **b** The 10 most frequently enriched biological process GO terms in nivolumab non-responders. Upregulated genes are functionally annotated using GO terminology using the R-package clusterProfiler. **c** Volcano plot of fold differences in genes between HCC samples from nivolumab responders ($n = 5$) and non-responders ($n = 5$). $P$ values were calculated using the Wald test. Secreted protein-coding genes belonging to the RNase superfamily or known to be associated with human cancer are shown in different colors. **d** Heat map of the expression of the most differentially expressed secreted proteins in (**c**) ($P < 0.001$; 34 upregulated and 10 downregulated genes). Data were analyzed using two-sided $t$-test. **e** OS and recurrence probability in HCC patients based on RNase1 expression. **f** Expression of secreted RNase1 protein in murine and human HCC cells. Representative results from 3 independent experiments. **g** Enzyme-linked immunosorbent assay (ELISA)

analysis of RNase1 expression in plasma samples from HCC patients ($n = 67$) and normal individuals ($n = 15$). **h** Correlation analysis of the plasma RNase1 concentrations and RNase1 expression levels in paired human HCC samples ($n = 55$; $R = 0.6$ [two-sided Pearson's chi-square test]). **i** OS and recurrence probability in HCC patients based on RNase1 plasma level. **j** IHC and IF staining of HCC samples from nivolumab responders ($n = 5$) and non-responders ($n = 8$). Left: Fluorescent multiplex immunohistochemical labeling using the indicated antibodies upon nivolumab-based treatment (upper panel) and immunohistochemical staining for RNase1 in human HCC samples (bottom panel). Scale bars, 50 μm (inset, 25 μm). Right: RNase1 immunohistochemistry scores for human HCC samples upon nivolumab exposure ($n = 13$; $P = 0.0216$). Results are presented as mean ± SD values. Statistical analysis: **g** two-sided Unpaired Student's $t$-test; **e** and **i** Log-rank test. Source data are provided as a Source Data file.

Next, we asked whether elevated RNase1 expression is involved in regulation of immunosuppression in TME. To answer this question, we performed cytometry by time of flight analysis of mouse HCC samples with validated antibodies (Supplementary Table 6). viSNE analysis suggested the existence of at least nine different populations of immune cells in the data set (Fig. 2f, g and Supplementary Fig. 2e, f). Of note, by analyzing the frequency of each subpopulation of immune subsets in control and RNase1-overexpressing tumors, we found a dramatic increase in the frequency of immunosuppressive myeloid subsets including, infiltrating macrophages (F4/80$^{hi}$), monocytes (Ly6C$^{hi}$), and neutrophils (Ly6G$^{hi}$) (Fig. 2h, i). Meanwhile, we observed significantly decreased percentages of dendritic cells (DCs; CD11c$^{hi}$), effector lymphocytes, and B cells when RNase1 was highly secreted by tumor cells (Fig. 2h–k). In addition, we explored the association between RNase1 and immune cell infiltration in HCC orthotopic model. Consistent with the subcutaneous model, we found a significant increase in the infiltration of macrophages, monocytes, and neutrophils (Fig. 2l), whereas a decreased infiltration of DCs, B cells, CD4$^+$ T, and CD8$^+$ T cells in RNase1-overexpressing tumor (Fig. 2l, m), by comparing the cell number of each subpopulation of immune subsets in control tumors. Furthermore, we analyzed the population of iNOS$^+$CD206$^-$ TAM (M1-like TAM) and iNOS$^-$CD206$^+$ TAM (M2-like TAM) in mice bearing orthotopic HCA-1-Vec and HCA-1-RNase1 tumor, and we found a significantly decreased infiltration of M1-like TAM, whereas an increased infiltration of M2-like TAM in HCA-1-RNase1 tumors, than in HCA-1-Vec tumors (Fig. 2n–p). The abundance of immunosuppressive myeloid cells and low infiltration of antitumor effector T cells revealed a potent immunosuppressive role for RNase1 in the TME.

Because RNase1 is a functional ligand of the tyrosine kinase receptor ephrin A4 (EphA4) and induces EphA4 downstream signaling molecules, such as ERK, activation in different cancer cell types[23], we screened a panel of HCC cell lines to measure their EphA4 expression. We found that EphA4 was highly expressed in multiple HCC cell lines (Supplementary Fig. 2g), suggesting that RNase1-induced EphA4/ERK pathway activation in those cells. As we speculated, we observed exogenous RNase1-induced phosphorylation of EphA4 and ERK in HCA-1 and Hep3B cells upon RNase1 exposure (Supplementary Fig. 2h, i). Because researchers observed a positive correlation between MAPK pathway activity and PD-L1 expression in human cancer cells[32–34], we asked whether activation of EphA4/ERK signaling by RNase1 affects PD-L1 protein levels in HCCs. To determine this, we detected PD-L1 expression by flow cytometry analyses, we found that the surface expression of PD-L1 was significantly increased upon RNase1 treatment in HCA-1 cell (Supplementary Fig. 2j) and that knockdown of EphA4 decreased RNase1-induced phosphorylation of ERK and inhibited RNase1-induced PD-L1 expression in HCCs (Supplementary Fig. 2k, l). In addition, we observed higher PD-L1 protein expression in HCA-1/R1 tumors, than in HCA-1/Vec tumors (Supplementary Fig. 2m). Upregulation of PD-L1 and decreased percentage of dendritic cell (CD11c$^{hi}$), effector lymphocytes and B cells as well as recruitment of abundant

immunosuppressive immune cells in RNase1-overexpressing tumors suggested the importance of RNase1 in modulating immune balance in favor of immunosuppression.

## RNase1 directs macrophage polarization toward the protumorigenic phenotype

Because our results described above suggested that overexpression of RNase1 promotes HCC resistance to anti-PD-1 therapy, we next explored the function of RNase1 in tumor immunity. We first assessed the association between the RNase1 level and immune cell infiltration pattern in HCC patients in The Cancer Genome Atlas (TCGA) database. Spearman correlation analyses revealed that the mRNA level for RNase1 was most correlated with macrophage infiltration (Supplementary Fig. 3a and Supplementary Table 7). Macrophages exhibit two main phenotypes, antitumorigenic (M1-like) and protumorigenic (M2-like). Thus, we next classified the patients with high and low RNase1 transcript levels based on their transcriptional scores. We found that a high RNase1 transcript level was negatively correlated with M1-like TAM infiltration, but positively correlated with M2-like TAM infiltration (Fig. 3a, b). Further analyses revealed that RNase1 mRNA expression was strongly associated with gene transcripts that were highly expressed on the M2-like TAM surface, including CD209, CD163, and CD206 (Supplementary Fig. 3b–d), demonstrating a potential function of RNase1 in macrophage polarization in the TME.

In support of those findings, we induced human monocytes (THP-1) to differentiate into uncommitted macrophages and then polarized them to different phenotypes by lipopolysaccharide (LPS)/interferon (IFN)-γ, interleukin (IL)-4, or RNase1 as a stimulus, as LPS/IFN-γ and IL-4 are the major endogenous macrophage-activating factors for M1-like and M2-like macrophages, respectively[35]. RNA-sequencing results showed that RNase1-treated macrophages (M[R1]) had global changes in the transcriptome similar to those of IL-4-treated macrophages (Supplementary Fig. 3e). To further examine the function of RNase1 in macrophage polarization, we compared the expression of markers used to define M1 and M2 macrophages, including cell surface markers, chemokines, and cytokines, in macrophages. Compared to the untreated or LPS/IFN-γ–treated group, the sequencing results revealed a high similarity in transcriptome changes between IL-4-treated and RNase1-treated macrophages (Supplementary Fig. 3e). In addition, compared to the untreated group, the expression of M1 markers (CD86, CD80, HLA-DRA, and HLA-DRB1) was significantly increased in LPS/IFN-γ–treated macrophages, and M2 phenotypic markers (CD200R, MRC1, VSIG4, and FCER2) was upregulated in IL-4-treated or RNase1-treated macrophages M(R1) (Fig. 3c). Moreover, compared to untreated or LPS/IFN-γ–treated macrophages, the patterns of proinflammatory cytokine and chemokines expression in M(R1) were similar to those in IL-4–treated macrophages, that display an M2-like cytokine/chemokine profile (Supplementary Fig. 3f, g).

Given that M(R1) exhibit the M2-like macrophage phenotype, we next examined whether RNase1 modulates macrophage polarization.

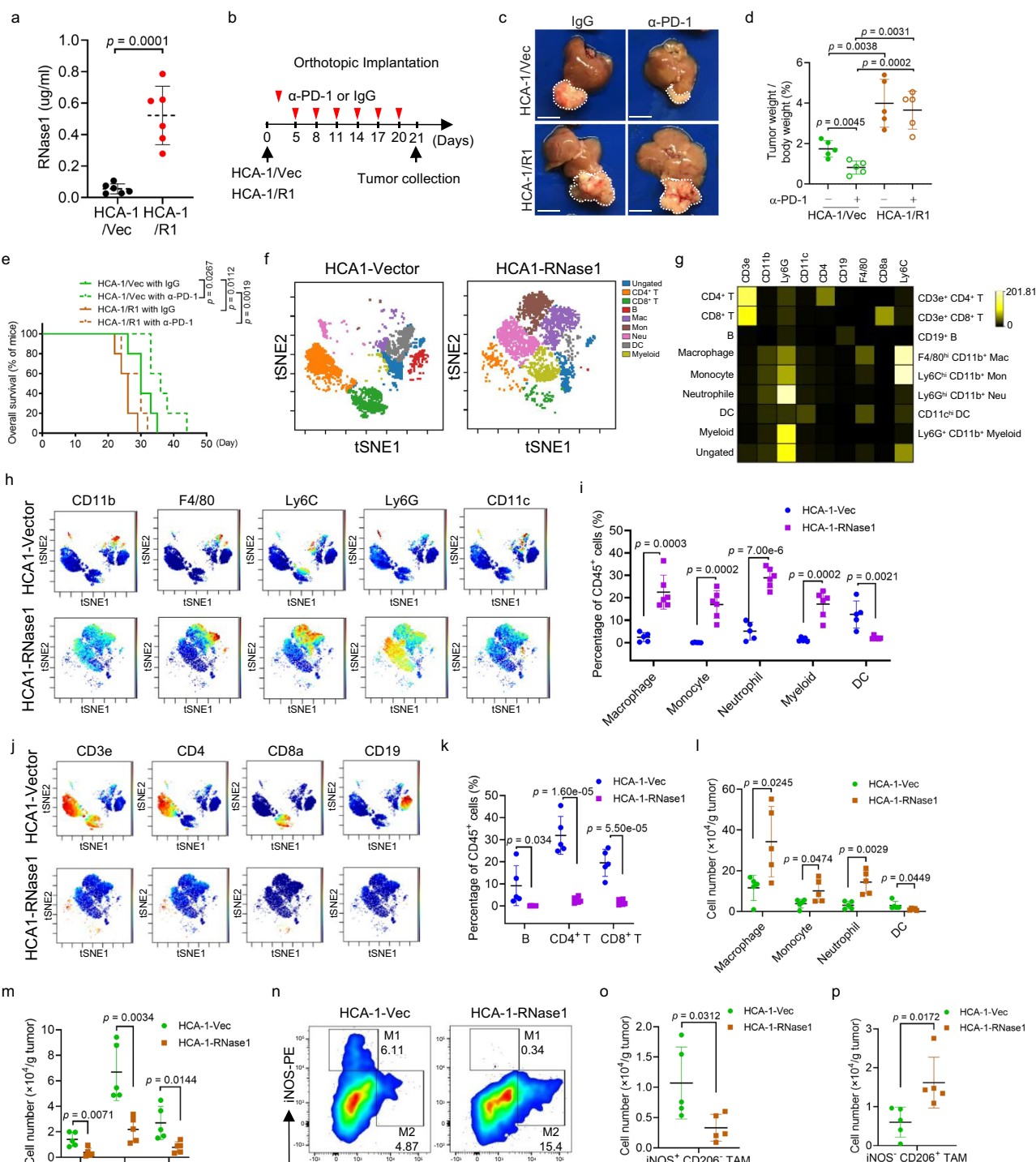

**Fig. 2 | RNase1 expression is correlated with extensive infiltration of immunosuppressive myeloid cells. a** The RNase1 plasma levels in mice bearing HCA-1/Vec or HCA-1/R1 tumors (*n* = 6 mice per group) on day 22 of treatment. **b** Schematic of the protocol for anti-PD-1 (αPD-1) and IgG-based treatment in the orthotopic HCC model. **c** Representative images of orthotopic liver tumors after αPD-1 or IgG treatment. **d** Normalized tumor weights measured at the treatment endpoint (*n* = 5 mice per group). **e** Survival of mice bearing HCA-1/Vec− or HCA-1/R1−derived orthotopic tumors following treatment with αPD-1 or IgG (*n* = 5 mice per group; log-rank test). **f** Identification of differentially distributed cellular phenotypes in HCA-1/Vec− and HCA-1/R1−derived tumor samples using the t-SNE algorithm. **g** Heat maps of the median immune cell marker intensities according to cytometry by time of flight-based immune profiling of HCA-1−derived tumors. **h** Myeloid cell populations

detected using CD11b, F4/80, Ly6C, Ly6G, and CD11c as markers. The cells in the map are color-coded according to the intensity of the expression of the indicated markers. **i** Quantification of the cell populations in (**h**). **j** Lymphocyte populations detected using CD3e, CD4, CD8, and CD19 as markers. **k** Quantification of the lymphocyte populations in (**j**). **f**–**k** *n* = 5 for HCA-1/Vec group and *n* = 6 for HCA-1/R1 group. **l** and **m** Cell number of myeloid cell (**l**) and lymphocyte (**m**) populations in orthotopic model. **n**–**p** Percentages (**n**) and absolute numbers (**o** and **p**) of iNOS⁺CD206⁻ and iNOS⁻CD206⁺ TAM subsets in the tumors. **l**–**p** *n* = 5 for HCA-1/Vec group and *n* = 5 for HCA-1/R1 group. Representative results from 3 independent experiments. The error bars represent (mean ± SD) values. Statistical analysis: **a**, **d**, **i**, **k**–**m**, **o**, **p** two-sided Unpaired Student's *t*-test. Source data are provided as a Source Data file.

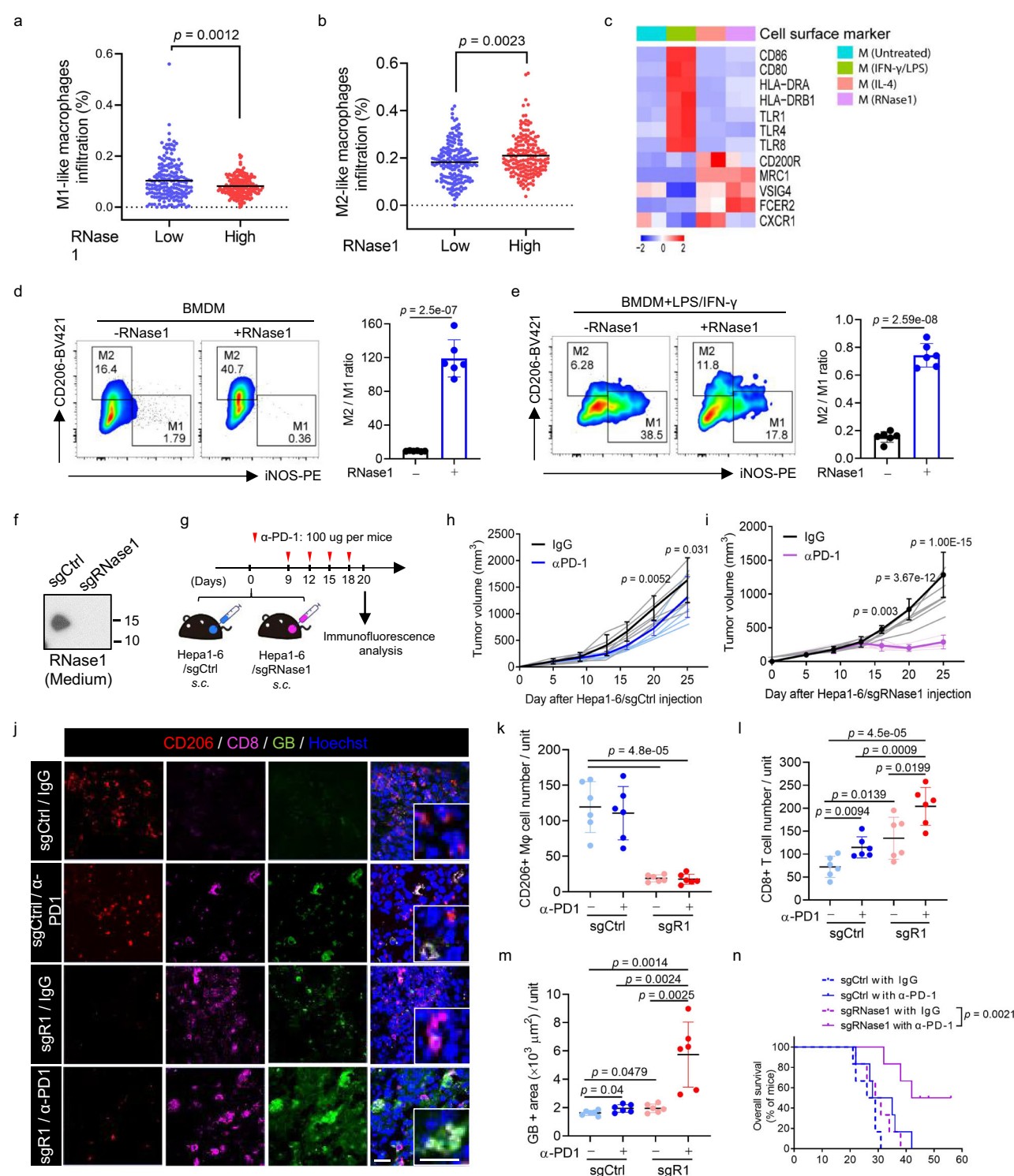

To that end, we generated bone marrow-derived macrophages (BMDMs) and treated them with recombinant RNase1 protein. We used the expression ratio for inducible nitric oxide synthase (iNOS; an M1 marker) (an M1 marker) versus CD206 (an M2 marker) to determine the M1-M2 polarization status, which generally denotes antitumor versus pro-tumor activities of TAMs[36]. The M2:M1 macrophage ratio in BMDMs was significantly increased upon RNase1 exposure (Fig. 3d). We found similar results for BMDMs treated with LPS/IFN-γ (Fig. 3e). Consistently, induction of RNase1 expression downregulated M1-like proinflammatory molecules (Nos2, Cxcl9, and Cxcl10) and de-repressed M2-like markers (Arg1 and Mrc1) in BMDMs

(Supplementary Fig. 3h). Taken together, these results implied that, educated by RNase1, macrophages were polarized into M2-like macrophages more potently.

To further determine whether RNase1-mediated macrophage polarization has a critical impact on anti-PD-1 therapy for HCC in vivo, we generated stable RNase1-deficient Hepa1-6 cells (Hepa1-6/sgRNase1) (Fig. 3f) and compared the response to treatment with αPD-1 between control Hepa1-6 wild-type (Hepa1-6/sgCtrl) and Hepa1-6/sgRNase1 cells using the same treatment schedule (Fig. 3g). Remarkably, mouse Hepa1-6/sgRNase1 tumors were much sensitive to αPD-1 than were mouse Hepa1-6/sgCtrl tumors as evidenced by a

**Fig. 3 | RNase1 regulates macrophage polarization and predicts reduced rates of response of HCC to anti-PD-1 therapy. a, b** RNase1 expression level was negatively correlated with M1-like macrophage infiltration (**a**) and positively correlated with M2-like macrophage infiltration (**b**). The M1- and M2-like macrophages are distinguished according to the average level of RNase1 expression in the TCGA database ($n = 178$ samples in both the RNase1-low and -high group). **c** Heat map of the expression of surface markers used to identify M1/M2 macrophages ($n = 2$ samples per group). **d** BMDMs were stimulated with or without RNase1 (1 µg/ml) for 24 h, stained for CD206 and iNOS, and analyzed using flow cytometry ($n = 6$ independent experiments per group). **e** BMDMs were stimulated with IFN-γ (10 ng/ml) plus LPS (200 ng/ml) in the presence or absence of RNase1 (1 µg/ml) for 24 h. The results of a representative experiment are shown in the left panel, and cumulative data from six independent experiments are shown in the right panel. **f** Immunoblot of secreted RNase1 in Hepa1-6 cells transduced with Cas9 and a control single guide RNA (sgCtrl) or sgRNase1. **g** Schematic of the treatment schedule for αPD-1 (100 µg per mouse) and IgG in C57BL/6 J mice injected with Hepa1-6/sgCtrl or Hepa1-6/sgRNase1 cells. Figure was created with Adobe Illustrator. **h, i** Tumor growth in mice bearing Hepa1-6/sgCtrl (**h**) or Hepa1-6/sgRNase1 (**i**) tumors after therapy with αPD-1 or IgG. The lightly colored lines represent individual tumor growth curves ($n = 6$ mice per group). **j** Representative images of immunostaining for CD206, CD8, and Granzyme B in mouse Hepa1-6 tumors. Hoechst: counterstaining. Scale bar, 20 µm. **k–m** Quantification of CD206 (**k**), CD8 (**l**), and Granzyme B (**m**) ($n = 6$ mice per group). Unit = 19,766 µm². **n** Survival rates for mice bearing Hepa1-6/Ctrl− or Hepa1-6/sgRNase1−derived tumors following treatment with αPD-1 or IgG ($n = 6$ mice per group; log-rank test). **f–h** Representative results from 3 independent experiments. The error bars represent (mean ± SD) values. Statistical analysis: **a, b, d, e, k, l,** and **m** two-sided Unpaired Student's $t$-test. **h** and **i** a log-rank test. Source data are provided as a Source Data file.

reduction in tumor growth (Fig. 3h, i). Consistent with this finding, immunofluorescent analysis showed significantly reduced CD206+ TAM and increased CD8+ cytotoxic T lymphocyte populations in mice bearing Hepa1-6/sgRNase1 tumors (Fig. 3j–l). Regarding T-cell activation, the effects of αPD-1 in mice bearing Hepa1-6/sgRNase1 tumors, as indicated by the levels of Granzyme B, appeared to be much better than those in mice bearing Hepa1-6/sgCtrl tumors (Fig. 3j, m). In addition, unlike in mice bearing Hepa1-6/sgCtrl tumors, treatment with αPD-1 significantly improved overall survival rates in mice bearing Hepa1-6/sgRNase1 tumors (Fig. 3n). These results demonstrated that depletion of tumor-secreted RNase1 reduced M2-like TAM populations, enhanced cytotoxic CD8+ T cells, as well as the efficacy of αPD-1−based immunotherapy for HCC.

## RNase1 promotes M2-like macrophage polarization through activation of ALK signaling

Our previous studies have identified that human RNases play a critical role in regulating several signaling activations, which was related to the oncogenic processes in human cancers[21,22]. Thus, we are interested in whether there has a relationship between RNase1 and oncogenic signatures. To address this, we performed RNA-seq and gene set enrichment analysis (GSEA) with THP-1 monocytes to see which gene sets were associated with RNase1 stimulation. We identified the five most prominently enriched oncogenic signatures under RNase1 or IL-4 stimulus (normalized enrichment score, >1.8; $P < 0.001$) (Supplementary Fig. 4a, b). Among those signatures, gene sets in the expression signature for ALK were significantly enriched in M(R1) but not IL-4−treated macrophages (Supplementary Fig. 4c). Because studies have verified that RNase family proteins can act as ligands to trigger receptor tyrosine kinase activation[21,22], and ALK is a receptor tyrosine kinase belonging to the insulin receptor superfamily[37], we next determined whether RNase1 promotes M2-like polarization by stimulating ALK signaling. Immunoprecipitation and Duolink assay results supported an association between RNase1 and ALK in ALK-expressing macrophage lines (Fig. 4a, b, Supplementary Fig. 4d) as well as EphA4-silenced BMDMs (Supplementary Fig. 4e). Furthermore, we observed enhanced phosphorylation of ALK in cells expressing WT ALK but not in cells expressing kinase-dead (I1250T) ALK (Fig. 4c). In addition, signal transducer and activator of transcription 3 (STAT3) signaling, a downstream signaling of the ALK pathway[38], was activated under RNase1 stimulus, as we detected dramatically increased phospho-ALK and phospho-STAT3 levels (Fig. 4d). Moreover, RNase1 exposure led to obvious reductions in the M1-like macrophage-related iNOS expression (Fig. 4e, Supplementary Fig. 4f) and STAT1 phosphorylation (Fig. 4e).

Because the balance between activation of STAT1 and STAT3 tightly regulates macrophage polarization and activity[35], our findings demonstrated that RNase1 may be a key player in macrophage polarization modulation through activation of ALK/STAT3 signaling. To confirm this, we investigated the function of RNase1 in macrophages after knocking down ALK in them using small interfering RNAs (siRNAs) or treatment with an ALK inhibitor (ALKi). ALK knockdown resulted in markedly lower phospho-ALK and phospho-STAT3 expression (Supplementary Fig. 4g) and dramatically higher iNOS and phospho-STAT1 expression (Fig. 4f) upon RNase1 exposure than that in cells with ALK expression. Moreover, ALK ablation restored Cxcl9 and Cxcl10 mRNA expression and decreased Arg1 and Mrc1 expression induced by the RNase1 stimulus (Fig. 4g, h), demonstrating that RNase1's effects on macrophages are mediated by ALK.

Next, we investigated the role of the RNase1/ALK axis on macrophage polarization after treatment with the ALKis crizotinib and alectinib, which are approved by the U.S. Food and Drug Administration for use in patients whose tumors exhibit increased ALK activity[38,39]. Treatment with either ALKi abrogated RNase1-induced M2-like polarization as evidenced by an increased iNOS+ macrophage population (Fig. 4i, j), upregulated iNOS (Fig. 4k), and decreased ALK and STAT3 activation (Fig. 4k). These data demonstrated that RNase1-triggered ALK activation is an approach to reprogram macrophage functionality by inhibiting M1-like macrophage polarization while promoting M2-like macrophage polarization.

## The combination of an ALKi and PD-1 blockade effectively relieves immunosuppression and suppresses tumor growth

Given that ALK blockade polarized RNase1-treated macrophages toward an M1-like phenotype (Fig. 4i–k) and RNase1 upregulated PD-L1 in HCC cells (Supplementary Fig. 2j), we asked whether combined treatment with an ALKi and αPD-1 can control cancer cells in vivo in the context of a complex TME. To answer this question, we administered crizotinib, αPD-1, or both to C3H mice bearing orthotopic HCA-1/R1 tumors (Fig. 5a). Crizotinib administration via oral gavage slightly reduced tumor sizes, which was similar to the findings in mice given αPD-1 alone. However, the combined therapy significantly improved tumor burdens in the mice (Fig. 5b, c). Importantly, treatment with crizotinib and αPD-1 did not change the body weights of the mice (Supplementary Fig. 5a) or cause elevated serum levels of aspartate aminotransferase, alanine transaminase, blood urea nitrogen, or creatine kinase (markers of liver and kidney damage) (Supplementary Fig. 5b). Notably, all vehicle-, crizotinib-, and αPD-1−treated mice had detectable metastasis, whereas three of five mice had much fewer metastases than did the other three groups of mice after the combination treatment (Supplementary Fig. 5c, d).

Consistent with the aforementioned mechanistic findings, flow cytometric analysis showed that CD206+ TAM populations were considerably smaller but that CD206− TAM populations were larger in tumors after treatment with crizotinib than in vehicle-treated tumors and that αPD-1 alone had no effect on TAM populations (Fig. 5d–f). The infiltrated populations of neutrophils and DCs are not significantly changed upon crizotinib or αPD-1 treatment (Supplementary Fig. 5e, f).

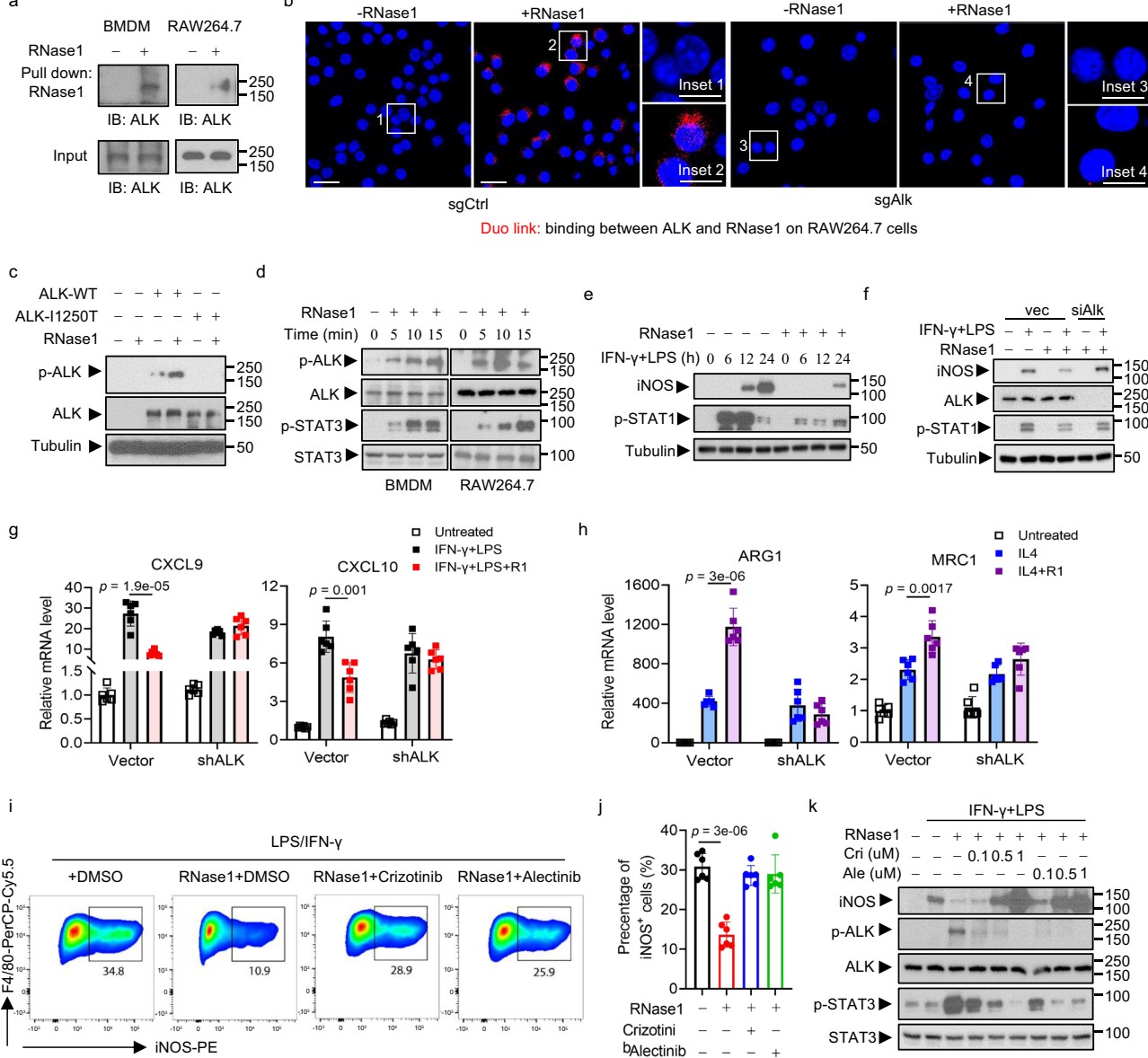

**Fig. 4 | RNase1 modulates macrophage polarization through ALK. a** Pull-down assay and Western blot analysis of the interaction between RNase1 and ALK in BMDMs and Raw264.7 cells. **b** Raw264.7 sgCtrl or sgAlk cells were treated with or without RNase1 (1 μg/ml) for 15 min. Detection of ALK and RNase1 binding was performed using a Duolink assay. Scale bar, 20 μm. **c** HEK-293T cells were transfected with plasmids containing a control vector (Vec), wild-type ALK (ALK-WT), or a kinase-dead mutant ALK (ALK-I1250T). An immunoblot of the indicated cells treated with RNase1 (1 μg/ml) for 15 min is shown. **d** Western blot of BMDMs and Raw264.7 cells treated with recombinant RNase1 protein (1 μg/ml) at various time points using the indicated antibodies. **e** BMDMs were treated with IFN-γ (10 ng/ml) plus LPS (200 ng/ml) for various time points in the presence or absence of RNase1 (1 μg/ml). **f** BMDMs transfected with ALK siRNAs and treated with or without IFN-γ plus LPS in the presence or absence of RNase1 for 24 h. **g**, **h** THP-1–derived

macrophages expressing ALK short hairpin RNA (shALK) were treated with IFN-γ plus LPS (**g**) or IL-4 (**h**) for 12 h in the presence or absence of RNase1. The indicated mRNA expression in these cells was measured using quantitative polymerase chain reaction and is shown as average values from six independent experiments. The error bars represent (mean ± SD) values. **i**, **j** BMDMs were pretreated with an ALKi for 2 h followed by IFN-γ plus LPS and RNase1 (1 μg/ml) for 24 h. The results of one representative experiment are shown in (**i**), and cumulative data from six independent experiments are presented in (**j**). The error bars represent mean ± SD values. **k** BMDMs were pretreated with various doses of crizotinib (Cri) or alectinib (Ale) for 2 h and then activated with IFN-γ plus LPS and RNase1 for 24 h (n = 6 mice per group). **a**–**f**, and **k** Representative results from 3 independent experiments. Statistical analysis: **g**, **h**, and **j** two-sided Unpaired Student's t-test. Source data are provided as a Source Data file.

Moreover, treatment with crizotinib or αPD-1 alone promoted greater numbers of tumor-infiltrating CD8[+] T cells (Fig. 5g, Supplementary Fig. 5g), IFN-γ[+]granzyme B[+]CD8[+] T cells (Fig. 5h), and T effector (Teff) subsets (Th1) of CD4[+] T cells (Fig. 5i, Supplementary Fig. 5h) in tumors than did treatment with the vehicle but substantially fewer than did the combination treatment (Fig. 5g–i). Furthermore, the numbers of regulatory T cells (Tregs; IFN-γ[−]Foxp3[+]) within the tumor-infiltrating CD4[+] T-cell population were dramatically reduced in the combination group

(Fig. 5j, Supplementary Fig. 5h). Notably, the combination treatment produced a significant improvement in the overall survival rate (Fig. 5k). Additionally, immunohistochemical staining demonstrated decreased expression of CD206 within tumors after crizotinib-based treatment, an association of induction of granzyme B expression with the combination therapy (Fig. 5l, m), and that the PD-L1 level was not affected after treatment with crizotinib, αPD-1, or both (Fig. 5m). These findings demonstrated that crizotinib enhances the efficacy of anti-PD-

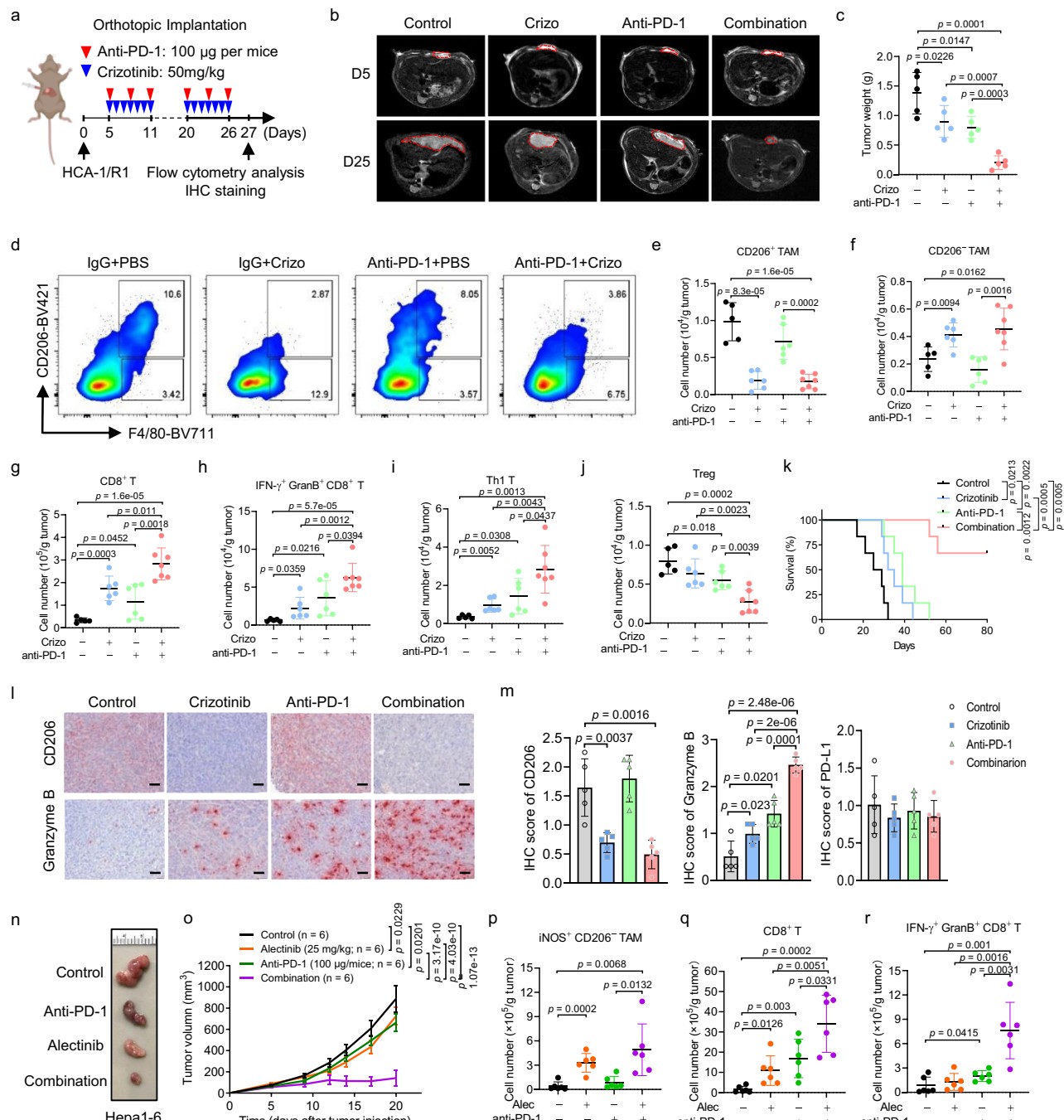

**Fig. 5 | The combination of an ALKi and anti-PD-1 therapy leads to macrophage polarization reprogramming and enhancement of immune effector functions.** **a** Schematic of the protocol for combination treatment with αPD-1 and crizotinib using an orthotopic HCC model. Figure was created with BioRender.com. **b** Magnetic resonance images of orthotopic liver tumors in the indicated groups of mice on day 5 (D5) and day 25 (D25) after tumor-cell implantation. **c** Mouse tumor weights measured at the treatment endpoint (*n* = 5 per group). **d–f** Percentages (**d**) and absolute numbers (**e**, **f**) of CD206⁺ and CD206⁻ TAM subsets in the tumors. **g, h** Absolute numbers of CD8⁺ T cells (**g**) and IFN-γ⁺granzyme B⁺ CD8⁺ T cells (**h**) in the tumors. **i, j** Absolute numbers of Th1-cell (**i**) and Treg (**j**) subsets in the tumors. For **e–j** *n* = 5 for control group; *n* = 6 for anti-PD-1 or Crizotinib treatment group; and *n* = 7 for combination group. **k** OS durations in mice bearing HCA-1-RNase1

tumors following treatment, as indicated (*n* = 6 mice per group). Log-rank test. **l** IHC staining from tumors of indicated group. Scale bar, 50 μm. **m** Quantification of IHC score of CD206, Granzyme B and PD-L1 in indicated groups (*n* = 5 per group). **n, o** The tumor growth of Hepa1-6 cells in C57BL/6 mice treated with Alectinib, anti-PD-1, or the combination. Representative images of tumors are shown on (**n**) and quantification of tumor volume is shown on (**o**). *n* = 6 mice per group. **p–r** Absolute numbers of iNOS⁺ CD206⁻ TAM (**p**), CD8⁺ T cells (**q**) and IFN-γ⁺ GranB⁺ CD8⁺ T (**r**) cells in the tumors (*n* = 6 per group). **a–k** Representative results from 2 independent experiments. **n–f** Representative results from 3 independent experiments. The error bars represent mean ± SD values. Statistical analysis: **c**, **e–j**, **m**, and **p–r**, two-sided Unpaired Student's *t*-test. **o** a log-rank test. Source data are provided as a Source Data file.

1 therapy by reprogramming macrophage polarization as well as inducing intertumoral accumulation of cytotoxic CD8+ and Th1 T cells.

Moreover, we also compared the combination of anti-PD-1 and Alectinib, another highly selective inhibitor of ALK, and single agent therapy in a subcutaneous Hepa1-6 liver cancer model (Supplementary Fig. 5i). The combination of Alectinib and PD-1 antibody also improved tumor growth inhibition in the subcutaneous model (Fig. 5n, o). In mice given Alectinib alone, the cell number of iNOS+ CD206- TAM was increased, whereas the cell number of iNOS- CD206+ TAM was decreased in the tumors (Fig. 5p and Supplementary Fig. 5j, k). Furthermore, the combination therapy also increased the population of CD8+ T-cell and IFN-γ+ GranB+ CD8+ T cells (Fig. 5q, r), which is consistent with the earlier results in the orthotopic model. These findings indicated that ALKi enhance efficacy of anti-PD-1 therapy by reprogramming macrophage polarization, as well as intertumoral accumulation of cytotoxic CD8+ T cells. Next, we examined the effects of ALKi on immunotherapy by establishing HCA-1 model. We found that anti-PD-1 treatment significantly decreased tumor size, whereas Alectinib has little effect on the inhibition of tumor growth. In addition, the efficacy of combination therapy was similar to that of anti-PD-1 alone, suggesting that ALK inhibitor has no off-target effects on RNase1-negative tumors (Supplementary Fig. 5l, m).

### Treatment with ALKi and PD-1 blockade result in accumulation of T cells with a memory phenotype

We next rechallenged the mice described above exhibiting complete HCA-1/R1 tumor regression after combination therapy with crizotinib and an anti-PD-1 antibody (Fig. 6a). Notably, the tumors were eradicated in five of six mice reinoculated with HCA-1/R1 cells, which was accompanied by a quite increased overall survival rate, whereas tumors grew rapidly in naïve mice inoculated in parallel with the same numbers of cells (Fig. 6b, c). Flow cytometry revealed dramatically more cytotoxic IFN-γ+CD8+ T, Th1, and Th17 cells in tumor-draining lymph nodes (TdLNs) in rechallenged mice than in tumor-free and tumor-bearing control mice. However, the numbers of Th2 cells and Tregs were comparable in these mice (Fig. 6d, Supplementary Fig. 6a−c). Consistently, significantly increased numbers of IFN-γ+CD8+ T and Th1 cells were generated within the spleen in rechallenged mice (Fig. 6e), demonstrating that combination therapy with an ALKi and αPD-1 elicited a long-term antitumor T-cell response.

To determine whether long-term immunity was established through generation of memory T (Tmem) cells in rechallenged mice, we analyzed the frequency of central memory ($T_{CM}$), effector memory ($T_{EM}$), and naïve T cells in TdLNs from tumor-free and tumor-bearing control and rechallenged mice. Rechallenged mice had markedly fewer $T_{CM}$ (CD62L+CD44+) and $T_{EM}$ (CD62L-CD44+) cells, with concomitant lower numbers of naïve (CD62L+CD44-) CD4+ and CD8+ T cells, than did the other two mouse groups (Fig. 6f, g). Authors recently reported that tissue-resident memory T (Trm) cells, based on their expression of CD69, represent a significant fraction of $T_{EM}$ cells, which affords robust protection against cancer progression through continuous surveillance of nonlymphoid tissues[40]. Therefore, we next examined the population of CD69+CD103+ and CD69+CD103- cells that resembled nonlymphoid Trm cells or displayed a profile that was more closely related to hat of recirculating cells[41]. We found that the numbers of both CD69+CD103+ and CD69+CD103- subsets of CD4+ and CD8+ Tmem cells were dramatically increased in TdLNs isolated from rechallenged mice (Fig. 6h, i, Supplementary Fig. 6d, e), suggesting that Trm cells were enriched and expanded upon combination treatment with an ALKi and αPD-1.

Circulating CD8+ T cells are heterogeneous[42], and the effector phase of infection predominantly consists of terminally differentiated effector cells, including short-lived effector cells (CD127-KLRG1+ Teff cells) and relatively few multipotent memory precursor effector cells (CD127+KLRG1- Tmem cells)[43,44]. Therefore, we further compared the populations of short-lived effector and memory precursor effector cells in each group of mice described above and found more memory precursor effector cells isolated from rechallenged mice than from tumor-free and tumor-bearing control mice, suggesting that combination therapy with an ALKi and αPD-1 can give rise to long-lived CD4+ and CD8+ Tmem cells (Fig. 6j, k, Supplementary Fig. 6f, g). In addition, the numbers of short-lived effector cells within TdLNs isolated from rechallenged mice were greater than those in TdLNs from the other mouse groups (Fig. 6j, k, Supplementary Fig. 6f, g). Taken together, these results demonstrated that a dynamic differentiation process occurs in CD4+ and CD8+ T cells from rechallenged mice, resulting in the formation of short-lived effector cells and memory precursor effector cells from early effector cells with a CD127-KLRG1- phenotype.

### Pathological relevance of RNase1 expression and immune status in HCC patients

Given that ALK is a receptor tyrosine kinase that is constitutively activated in patients with certain cancers[45], we asked whether RNase1 induces ALK activation in HCC patients. To answer this, we first screened a panel of HCC cell lines for the expression of ALK and found that it was highly expressed in most HCC cells (Supplementary Fig. 7a). Of note, RNase1 expression was positively correlated with phospho-ALK expression in HCC tissue sections (Supplementary Fig. 7b, c), suggesting a potential oncogenic role for RNase1 in ALK-expressing HCCs. Next, to determine the correlation between RNase1 expression and infiltration of TAMs and cytotoxic T lymphocytes in patients with HCC, we evaluated the correlation among RNase1, CD68, CD206, and CD8 expression in the HCC cohort. We observed that RNase1 expression was positively correlated with CD68 ($P = 0.004$) and CD206 ($P = 0.016$) expression (Fig. 7a, Supplementary Fig. 7d, e), supporting the pathological relevance of RNase1 and CD206+ TAM infiltration. Moreover, we evaluated the level of cytotoxic T-cell infiltration using immunohistochemistry with antibodies against CD8. We found that infiltration of CD8+ cells was negatively associated with RNase1 expression ($P = 0.024$) (Fig. 7b, Supplementary Fig. 7f). In addition, we observed a significant association between RNase1 expression and PD-L1 expression in tumor sections ($P = 0.003$) (Fig. 7c, Supplementary Fig. 7g). Taken together, the positive correlations among RNase1 expression, CD206+ TAM infiltration, and PD-L1 expression and negative correlation between RNase1 expression and CD8+ T-cell infiltration in tumor sections (representative cases shown in Fig. 7d) supported that RNase1 is a potential biomarker for predicting immunosuppressive TMEs in HCC patients. Also, our finding that increased RNase1 expression was accompanied by M2-like macrophage polarization through the RNase1/ALK axis suggested that treatment with an ALKi in combination with an anti-PD-1 antibody decreases immunosuppression in HCC patients with high RNase1 expression (Fig. 7e).

## Discussion

Clinical research focused on the predictive capacity of biomarkers in evaluating tumor response to immunotherapies has implicated some serum biomarkers to be prognostic factors associated with good outcomes of HCC, such as PIVKA, DCP, and VEGF[46–48]. However, no identified biomarkers have been able to accurately identify the best candidates for immunotherapy. One of the main explanations is that the TME is highly heterogeneous in the liver[49,50]. In the present study, we identified RNase1 as one of the most upregulated secreted proteins in non-responders receiving nivolumab (Fig. 1d) and that RNase1 expression is associated with high abundance of TAMs or neutrophils in the TME (Fig. 2h, i), which are generally associated with poor prognosis for and immunotherapy resistance of many types of cancer[49,51]. We understand the size of cohort is small, but the results are encouraging, and potentially important therefore we performed the following laboratory experiments to provide additional support on the concept. Notably, by analyzing public datasets derived from gastric cancer ($n = 45$, ERP107734, anti-PD-1 treatment) and renal

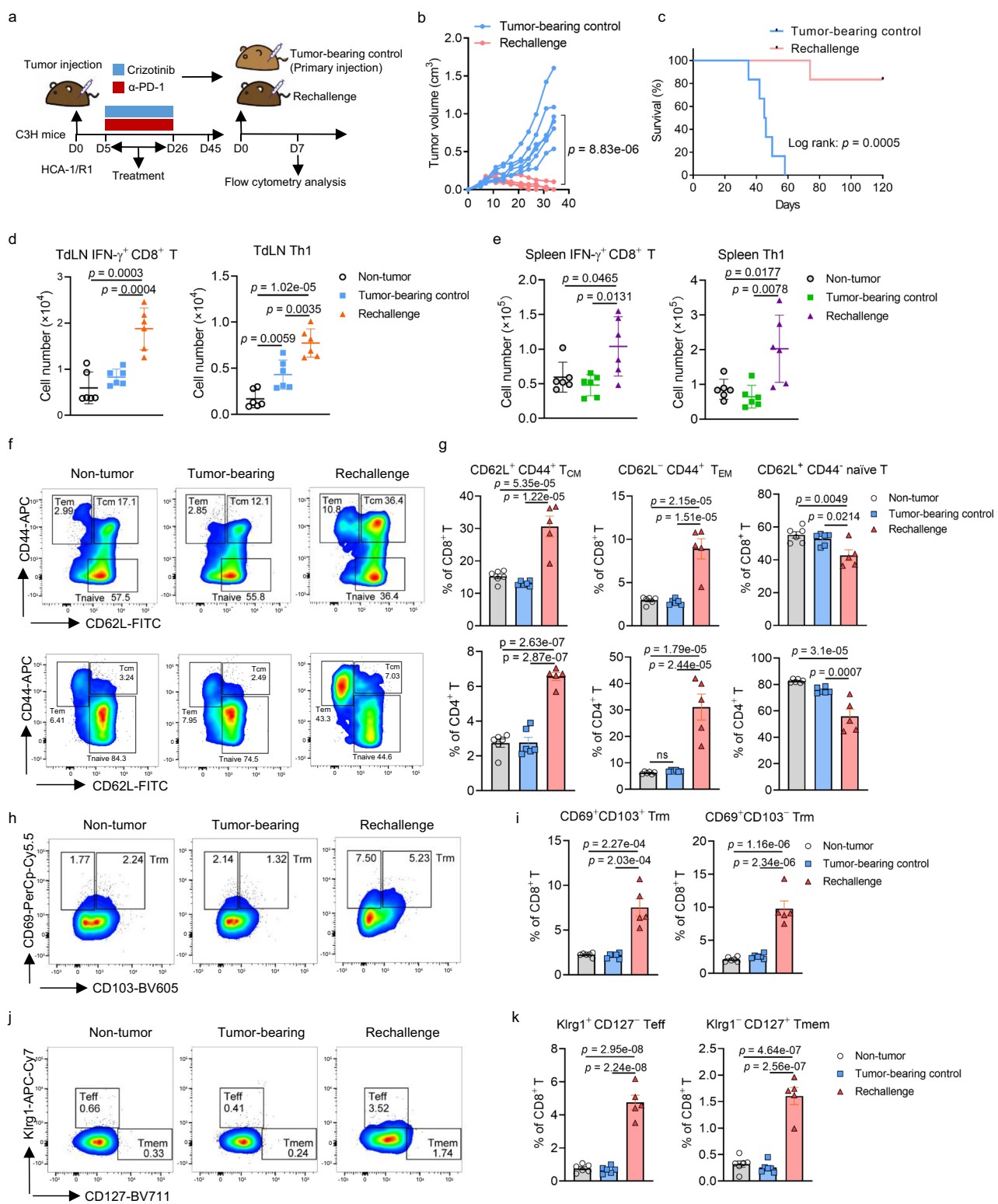

cell carcinoma (RCC) datasets (*n* = 29, SRP128156, anti-PD-1/anti-PD-1+anti-CTLA4 treatment) through "Immune Checkpoint Blockade Therapy Atlas (ICBatlas, URL: http://bioinfo.life.hust.edu.cn/ICBatlas/#!/), we found that non-responders displayed significantly higher expression of RNase1 than responders in gastric cancer and RCC datasets (*P* = 0.045 and *P* = 0.010) (Supplementary Table 8). Our findings may open a direction in our understanding of human RNases and provide a potential plasma biomarker of response to immunotherapy for cancer.

Consistent with previous studies demonstrating the ligand function of RNase1 in EphA4 in breast cancer cells[23], our data demonstrated that RNase1 phosphorylated EphA4 and ERK1/2 in two HCC cell lines (Supplementary Fig. 2h, i). A recent study showed that activation of the MAPK/ERK pathway increased PD-L1 mRNA and protein levels in lung adenocarcinoma cells[32]. Moreover, an activated NRAS/RAF/MEK1/2/ERK/c-Jun axis enhanced the transcription of PD-L1 in melanoma cells[52], and induced PD-L1 expression by activating the EGFR/MAPK pathway in pancreatic cancer cells[53]. The results of our flow cytometry

**Fig. 6 | Treatment with an ALKi and anti-PD-1 antibody results in protection from a secondary tumor challenge in an HCC mouse model. a** Scheme of the experimental procedure for the HCA-1/R1 tumor regression study. C3H mice were challenged with stable HCA-1/R1 cells and given crizotinib and an anti-PD-1 antibody as indicated in Fig. 5a. Mice with complete tumor regression were rechallenged 45 days after tumor-cell injection (*n* = 12). Similarly aged C3H mice (*n* = 12) were injected with HCA-1/R1 cells and used as tumor-bearing controls. TdLNs and splenocytes were harvested from mice for flow cytometric analysis 7 days after tumor rechallenge (*n* = 6 per group). Figure was created with Adobe Illustrator. **b** The sizes of the tumors over time in the rechallenge and tumor-bearing control groups (*n* = 6 per group). **c** The overall survival rates for the rechallenged and tumor-bearing control mice (*n* = 6 per group). **d, e** The numbers of CD8$^+$IFN-γ$^+$ T cells and Th1 cells in the TdLNs (**d**) and spleens (**e**) of the indicated groups of mice (*n* = 6 per group). The error bars represent mean ± SD values. **f, g** The percentages (**f**) and absolute numbers (**g**) of CD8$^+$ and CD4$^+$ T$_{CM}$, T$_{EM}$, and naïve T-cell subsets in TdLNs. **h, i** The percentages (**h**) and absolute numbers (**i**) of CD69$^+$CD103$^+$ and CD69$^+$CD103$^-$ subsets of CD8$^+$ Trm cells in TdLNs. **j, k** Percentages (**j**) and absolute numbers (**k**) of Klrg1$^+$CD127$^-$CD8$^+$ Teff cells and Klrg1$^-$CD127$^+$CD8$^+$ Tmem cells in TdLNs. For **g, i** and **k:** *n* = 6 for non-tumor and tumor-bearing control groups; *n* = 5 for rechallenge group. The error bars represent (mean ± SEM) values. **a–c** Representative results from 2 independent experiments. The results were analyzed using two-way analysis of variance (ANOVA; **b**), a log-rank test (**c**), a two-sided unpaired Student *t*-test (**d, e**), or one-way ANOVA (**g, i**, and **k**).

analysis of PD-L1 expression in RNase1-treated HCC cells were in agreement with findings in those previous studies and indicated a mechanism by which RNase1 induces PD-L1 expression by activating the EphA4/ERK pathway in HCC cells (Supplementary Fig. 2j–l).

ALK is primarily known for its oncogenic role in several human cancers, and it is a therapeutic target for ALK-altered cancers[54]. Results of immunohistochemical staining of tumor samples from HCC patients demonstrated a correlation between RNase1 expression and phospho-ALK expression in cancer cells (Supplementary Fig. 7b, c) and that ALK was expressed in most HCC cell lines (Supplementary Fig. 7a). Interestingly, we observed increased viability and invasion in Hep3B and Huh7 cells upon RNase1 treatment, which could be repressed by pretreatment with crizotinib or Alectinib (Supplementary Fig. 7h–j), suggesting that ALK is also essential for the biological activities of RNase1 in HCC tumor cells. Therefore, in HCC patients with high expression of RNase1, treatment with an ALKi may have dual effects in that it not only contributes to modulation of macrophage phenotypes but also suppresses growth of tumors harboring ALK expression. Indeed, our data demonstrated that treatment with an ALKi alone decreased CD206$^+$ TAM infiltration and reduced tumor weights in the RNase1-expressing orthotopic HCC model (Fig. 5c–e). Notably, the combination of an ALKi and anti-PD-1 therapy had markedly better antitumor effects than did anti-PD-1 treatment alone without any detectable toxic effects and resulted in the accumulation of T cells with a long-memory phenotype (Figs. 5 and 6), suggesting that ALKi has strong potential for use in combination with immunotherapy in HCC patients with high RNase1 expression.

Various preclinical and clinical studies have identified new roles for ALK in immune evasion as well as innate, humoral, and T-cell immunity[54]. For example, by employing a sepsis model and subsequent silencing of ALK gene in immortalized mouse BMDMs, Zeng et al first demonstrated ALK's involvement in innate immunity against microbial pathogens[55]. We explored a role for ALK as a modulator of macrophage polarization in response to exposure to RNase1. Mechanistically, RNase1 binds to and activates ALK to promote STAT3 phosphorylation, which then attenuates expression of M1-like proinflammatory molecules (Nos2, Cxcl9, and Cxcl10) and increases expression of M2-like markers (Arg1 and Mrc1) (Fig. 4). We later found this RNase1/ALK axis to be associated with immunotherapy response in mouse HCC models, in which crizotinib- or alectinib-mediated inhibition of ALK activity significantly reduced resistance to anti-PD-1 therapy. We believed a reduction in the number of deaths and reduced rate of tumor growth in ALKi and anti-PD-1 combination treatment group partially result from reduction of CD206$^+$ TAM infiltration and increased cytotoxic T-cell recruitment (Fig. 5). Interestingly, recent results of Wang et al., who found that RNase1 elicits adaptive immune response against breast cancer by boosting CD4$^+$ T-cell activation through associating with EphA4[56]. The different binding receptors of RNase1 on macrophage and CD4$^+$ T cells might cause the differential responses in the two cell and cancer types. As we know, the number of TAMs in the TME accounts for 20% to 40% among all HCC infiltrating lymphocytes and even more in some rare HCC subtypes[57]. Thus, RNase1 exhibits immunosuppressive effects in our HCC mouse models through

activating ALK on TAMs. RNase1 may regulate CD4$^+$ T cells activation through binding to different receptor, EPHA4, thus it would be worthwhile to evaluate the proportion of TAMs and CD4$^+$ TILs, as well as the expression level of ALK and EphA4 in different patients when these markers are used for choosing appropriate treatment strategy.

Our recent understanding of ALK in macrophage polarization and immune responses has begun to provide initial clues for the design of future ALK-based immunotherapy for HCC. Large-scale preclinical studies and clinical trials are warranted to determine whether RNase1/ALK axis inhibition and combinational immune checkpoint blockade therapy are potentially effective therapeutic strategies for all major cancer types. This report provides the scientific basis for clinically testing this possibility.

## Methods

### Study approval
Tumor samples for RNA-seq analyses, human HCC tissue microarray for IHC analyses and plasma samples of HCC patients and healthy individuals for ELISA analyses were obtained from Zhongshan Hospital affiliated to Fudan University. All tissue samples and plasma samples were collected and used in compliance with informed consent policy. The collection and the use of all samples were approved by the Zhongshan Hospital Research Ethics Committee and complied with all relevant ethical regulations. All animal-based research was conducted according to the guidelines and requirements set forth by the Public Health Service Policy on Humane Care and Use of Laboratory Animals, the U.S. Department of Health and Human Services Guide for the Care and Use of Laboratory Animals, the Animal Welfare Act, and the MD Anderson Institutional Animal Care and Use Committee.

### Cell culture and treatment
HEK 293 T cells (CRL-3216), Hep3B cells (HB-8064), HepG2 (HB-8065), Hepa1-6 (CRL-1830) and PLC/PRF/5 (CRL-8024) were purchased from American Type Culture Collection (ATCC). HCC cell lines Huh7, HCA-1, HA22T, HA59T, Tong, and Mahlavu were obtained from the China Medical University Hospital. The cell lines used in this study were authenticated by STR profiling and tested for the absence of myco-plasma contamination.

Mouse BMDMs were obtained as described previously[58] and maintained in Dulbecco's modified Eagle's medium supplemented with 10% fetal bovine serum and 10% supernatants of L929 mouse fibroblasts as a conditioned medium providing macrophage colony-stimulating factor. After 6 days of culture, floating cells were discarded, and attached macrophages were plated in multiple well plates overnight prior to stimulation. For a THP-1 monocyte polarization experiment, human THP-1 monocytes (ATCC; #TIB-202) were differentiated into macrophages with 200 nM phorbol myristate acetate (P8139; MilliporeSigma) for 48 h and then cultured with the addition of 10 ng/ml human IFN-γ plus 200 ng/ml LPS or 20 ng/ml human IL-4. Murine macrophage-like RAW264.7 (ATCC; #TIB-71) and HCC cell lines were cultured in Dulbecco's modified Eagle's medium supplemented with 10% fetal bovine serum and antibiotics. Unless specified

a

| | | RNase1 expression | | | |
|---|---|---|---|---|---|
| | | Low | High | Total | P value |
| **CD68** | Low | 52(61.9%) | 36(40%) | 88(50.6%) | |
| | High | 32(38.1%) | 54(60%) | 86(49.4%) | |
| Total | | 84(100%) | 90(100%) | 174(100%) | p=0.004 |
| **CD206** | Low | 58(69%) | 46(51.1%) | 104(59.8%) | |
| | High | 26(31%) | 44(48.9%) | 70(40.2%) | |
| Total | | 84(100%) | 90(100%) | 174(100%) | p=0.016 |

b

| | | RNase1 expression | | | |
|---|---|---|---|---|---|
| | | Low | High | Total | P value |
| **CD8** | Low | 37(44%) | 55(61.1%) | 92(52.9%) | |
| | High | 47(56%) | 35(38.9%) | 82(47.1%) | |
| Total | | 84(100%) | 90(100%) | 174(100%) | p=0.024 |

c

| | | RNase1 expression | | | |
|---|---|---|---|---|---|
| | | Low | High | Total | P value |
| **PD-L1** | Low | 58(69%) | 42(46.7%) | 100(57.5%) | |
| | High | 26(31%) | 48(53.3%) | 74(42.5%) | |
| Total | | 84(100%) | 90(100%) | 174(100%) | p=0.003 |

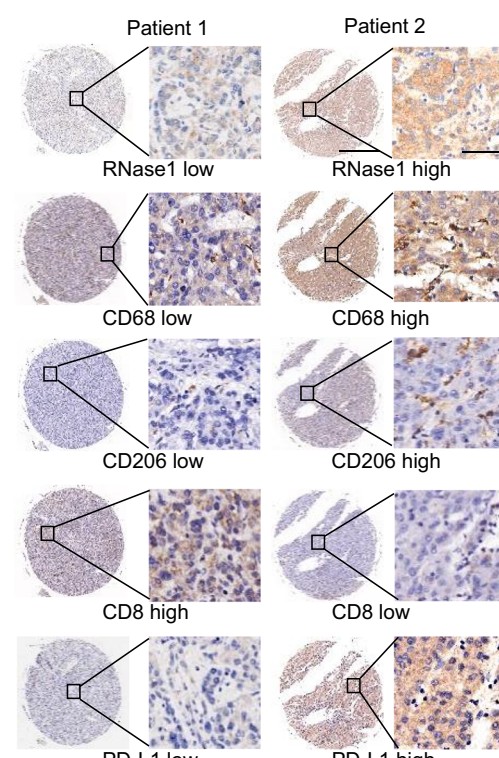

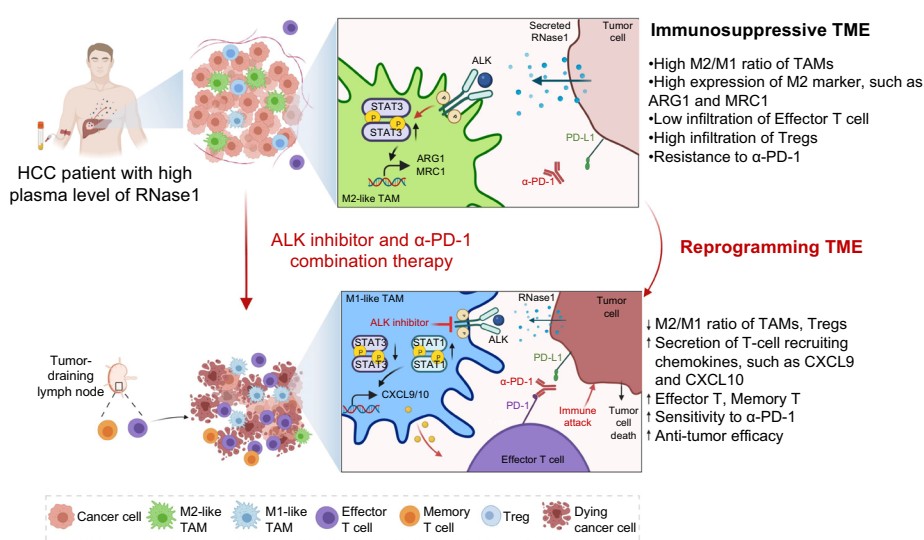

**Fig. 7 | Pathological relevance of RNase1, CD206, CD8, and PD-L1 expression in HCC patients. a–c** Quantification of immunohistochemical staining for the correlation between RNase1 and CD206 (**a**), RNase1 and CD8 (**b**), and RNase1 and PD-L1 (**c**) expression using a human HCC tissue microarray. Correlations were assessed using the two-sided Pearson chi-square test. A *P* value less than 0.05 was set as the criterion for statistical significance. **d** Two representative cases from the immunohistochemical staining in (**a–c**). Scale bar, 50 μm. **e** A proposed model of RNase1 as a secretory biomarker used to identify treatment options for HCC. In brief, high plasma levels of RNase1 induce immunosuppression by promoting the polarization of TAMs via binding to ALK and trigger ALK/STAT3 signaling, which in turn debilitates antitumor immune responses. ALK inhibitor treatment enhances the efficacy of anti-PD-1 therapy by reprogramming macrophage polarized from pro-tumor phenotype (M2-like) into antitumor phenotype (M1-like), inducing the secretion of T-cell recruiting chemokines, intertumoral accumulation of cytotoxic CD8⁺ T cells, as well as reduction of Tregs. In addition, memory T-cell were enriched and expanded in rechallenged mice. ALK activation and immunosuppressive state triggered by RNase1 could be reversed by ALK inhibitor and anti-PD-1 combination therapy, and RNase1 could serve as a plasma biomarker to identify patients with HCC who may benefit from this therapeutic strategy. Figure was created with BioRender.com.

otherwise, cells were treated with RNase1 at a concentration of 1 μg/ml under serum-starved conditions. The crizotinib and alectinib concentrations used in an in vitro treatment assay were 1 μM and 0.5 μM, respectively. Cell lines were validated using short tandem repeat (STR) DNA fingerprinting with an AmpFLSTR Identifiler kit (Applied Biosystems) according to the manufacturer's instructions. The STR profiles were compared with known ATCC fingerprints (https://www.ATCC.org), the Cell Line Integrated Molecular Authentication database (version 0.1.200808; Nucleic Acids Research 37: D925-D932 PMCID: PMC2686526), and The University of Texas MD Anderson Cancer Center fingerprint database. The STR profiles matched known DNA fingerprints or were unique.

## Stable cell line generation

Lentiviral-based plasmids pLV-RNase1-GFPSpark (#MG53784-ACGLN) for RNase1 overexpression was purchased from Sino Biological. pLV-C-GFPSpark lentivirus plasmid (#LVCV-35) was used as a control, and RNase1 CRISPR Guide RNA (target sequence: TGCCAAGGGCTCATG-CACGA) for RNase1 knockout in Hepa1-6 cells was purchased from GenScript. ALK CRISPR/Cas9 KO Plasmid (sc-419085-KO-2) was from Santa Cruz Biotechnology. A control CRISPR/Cas9 plasmid encoding a non-specific 20 nt guide RNA (sc-418922, Santa Cruz Biotechnology) was used as a negative control. A lentiviral-based short hairpin RNA used to knockdown ALK was purchased from Sigma. shALK#1 (TRCN0000000786; Target Sequence: ACCCAAATCAAGAAACCTGTT), and shALK#2 (TRCN0000000787; Target Sequence: AGAAGAA-GAAATCCGTGTGAA). For viral infection, the packaging plasmids VSV-G and dvpr were co-transfected with the desired genes into HEK-293T cells, and viral particles were harvested 72 h after transfection. HCC cells were infected with viruses for 8 h in the presence of hexadimethrine bromide (Polybrene; 10 µg/ml) and subsequently selected using flow sorting (HCA-1/R1) or puromycin (2 µg/ml; Hepa1-6/sgRNase1 and RAW264.7/sgALK). Lipofectamine 2000 (Life Technologies) was used for transient transfection.

## Antibodies, proteins, and reagents

The primary antibodies used for Western blotting, immunofluorescent analysis, and immunoprecipitation were rabbit anti-RNase1 (Polyclonal, cat. no. HPA001140; Atlas Antibodies; 1:1000) and mouse anti-tubulin (clone B-5-1-2, cat. no. T5168; Sigma-Aldrich; 1:3000); rabbit anti-phospho-ALK (Y1057, cat. no. ab192809; Abcam; 1:1000) and rat anti-CD8 (clone YTS169.4, cat. no. ab22378; Abcam; 1:100); mouse anti-EphA4 (clone M280, cat. no. EM2801; ECM biosciences; 1:1000) and anti-phospho-EphA4 (Y602, cat. no. EP2731; ECM Biosciences; 1:1000); rabbit anti-PD-L1 (clone 2096C, cat. no. MAB90781; R&D Systems; 1:100) and goat anti-granzyme B (cat. no. AF1865; R&D Systems; 1:100); mouse anti-ALK (clone F-12, cat. no. sc-398791; Santa Cruz Biotechnology; 1:1000) and mouse anti-STAT3 (clone F-2, cat. no. SC-8019; Santa Cruz Biotechnology; 1:2000); rabbit anti-phospho-STAT3 Tyr705 (clone D3A7, cat. no. 9145S; Cell Signaling Technology; 1:2000), rabbit anti-phospho-STAT1 (clone 58D6, cat. no. 9167; Cell Signaling Technology; 1:2000), rabbit anti-phospho-ERK1/2 Thr202/Tyr204 (Polyclonal, cat. no. 9101; Cell Signaling Technology; 1:3000), rabbit anti-ERK1/2 (Polyclonal, cat. no. 9102; Cell Signaling Technology; 1:3000), and rabbit anti-iNOS (Polyclonal, cat. no. 2977; Cell Signaling Technology; 1:1000); and rabbit anti-CD206 (Polyclonal, cat. no. NBP1-90020; Novus Biologicals; 1:100). The following antibodies were used for immunohistochemical analyses: goat anti-PD-L1 (Polyclonal, cat. no. AF1019; R&D Systems; 1:200), rabbit anti-RNase1 (Polyclonal, cat. no. HPA001140; Sigma-Aldrich; 1:500), rabbit anti-CD206 (Polyclonal, cat. no. NBP1-90020; Novus Biologicals; 1:500), rabbit anti-granzyme B (Polyclonal, cat. no. 4059; Abcam; 1:200), rabbit anti-phospho-ALK (Polyclonal, cat. no. 3341; Cell Signaling Technology; 1:100), rabbit anti-CD8 (clone SP16, cat. no. ab101500; Abcam; 1:100), and rabbit anti-CD68 (clone EPR20545, cat. no. ab213363; Abcam; 1:200). The following antibodies were used for the animal studies: 100 µg of a mouse anti-PD-1 antibody (cat. no. BE0146) and IgG control (cat. no. BE0089; Bio X Cell).

Human recombinant RNase1 protein was prepared using a procedure described previously[22]. Briefly, an RNase1 gene fragment with *Escherichia coli* codon bias was subcloned into the prokaryotic expression vector PSJ3 with an N-terminal 6 × His tag. *E. coli* BL21(DE3) cells carrying the desired plasmids were grown in Terrific Broth at 37 °C to an OD600 of 0.6–0.8. Protein expression was induced by the addition of 0.5 mM IPTG, and incubation was performed for 4 h. The target proteins were extracted from inclusion bodies, refolded, and purified using immobilized metal-affinity chromatography on an Ni-NTA column followed by size-exclusion chromatography (HiLoad Superdex 200; GE Healthcare). Recombinant mouse RNase1 protein (cat. no. abx068908) was obtained from Abbexa, and LPS (cat. no. sc-3535) was obtained from Santa Cruz Biotechnology. Human IL-4 (cat. no. 200-04) was obtained from PeproTech. Human IFN-γ (cat. no. 570202) was obtained from BioLegend. Mouse IL-4 (cat. no. 51084-MNAE) and mouse IFN-γ (cat. no. 50709-MNAH) were obtained from Sino Biological. The ALKi crizotinib (cat. no. PF-02341066) was purchased from Selleck Chemicals. The ALKi alectinib (cat. no. AF802) was obtained from LC Laboratories.

## Tissue samples

For the HCC immunotherapy patient cohorts, the tumor tissues and sample information were retrospectively collected from pretreatment HCC patients from Zhongshan Hospital, Fudan University, Shanghai, China. Those patients were standard treated with anti-PD-1 antibodies between May 2020 and January 2021 with the following criteria: (1) Clinically confirmed HCC based on the clinical criteria or histology; (2) Tumor was considered unresectable; (3) At least one measurable tumor lesion according to the modified Response Evaluation Criteria in Solid Tumors (mRECIST)[59]; (4) The clinical data of these patients could be obtained from medical record system in Zhongshan Hospital; (5) No history of other malignant cancers. For Fig. 1a, 10 tumor samples are randomly selected to do RNA-seq analysis based on the response assessment to anti-PD-1 treatment according to the mRECIST. For Fig. 1j, 13 cases of tumor tissue slides were stained for IHC or IF markers. Detailed clinicopathological features of the patients are listed in Supplementary Tables 1 and 5. HCC cohort consisted of 174 patients who underwent surgical resection and whose samples were collected and monitored in the Liver Surgery Department at Zhongshan Hospital. Paired plasma samples from HCC patients (*n* = 67) and healthy individuals (*n* = 15) were analyzed using an ELISA to measure RNase1 levels. The collection and the use of human HCC tissue samples, plasma samples from HCC patients and healthy individuals were approved by the Zhongshan Hospital Research Ethics Committee and complied with all relevant ethical regulations.

## In vivo murine models

All mice were housed in 12 h light/dark cycle with controlled room temperature (23 ± 2 °C) and humidity (30–70%). To generate the subcutaneous HCC model, C3H or C57BL/6 mice (male, 6–8 weeks old; The Jackson Laboratory) were subcutaneously injected with $1 × 10^6$ stable HCA-1 cells or $5 × 10^6$ stable Hepa1-6 cells as indicated. Tumor volumes were monitored using external calipers and calculated using the formula (length × width²)/2. Tumor-bearing mice in both groups were further placed into two groups for treatment with 100 µg of an anti-PD-1 antibody (RMP1-14; Bio X Cell) or IgG (control; Bio X Cell). Treatment was administrated intraperitoneally every 3 days starting 1 week after tumor-cell inoculation. To generate the orthotopic HCC model, subcutaneous HCA-1/Vec or HCA-1/R1 tumors were cut into cubes (1 mm³) under aseptic conditions. Next, single cubes were inoculated into the liver parenchyma of C3H mice that were anesthetized using isoflurane. The mice were further placed in treatment groups and given daily oral doses of 50 mg/kg crizotinib for 3 weeks (5 days a week), 100 µg of the anti-PD-1 antibody, or rat IgG intraperitoneally every 3 days starting on day 5 after implantation. For an HCA-1/R1 tumor regression study, $1 × 10^6$ stable HCA-1/R1 cells were inoculated into tumor-bearing mice with complete tumor regression after combined treatment with the anti-PD-1 antibody and crizotinib. And treatment naïve mice were inoculated with $1 × 10^6$ HCA-1/R1 cells in parallel. Subcutaneous tumors were measured using calipers, and orthotopic tumors were evaluated using a Bruker 7 T magnetic resonance imaging scanner. Mice was euthanized using $CO_2$ exposure followed by cervical dislocation when any direction of the tumor exceeds 1.5 cm, or ulceration happens, or at the experimental endpoint, and their tumors were excised for subsequent analysis. In some

cases, this limit has been exceeded the last day of measurement and the mice were immediately euthanized.

## Cytometry by time of flight and flow cytometry

For cytometry by time of flight analysis, HCA-1/Vec and HCA-1/R1 tumors were excised and digested to single cells using a gentleMACS Dissociator and a mouse Tumor Dissociation Kit (Miltenyi Biotec). Tumor-infiltrating lymphocytes were enriched on a Ficoll gradient (Sigma-Aldrich), incubated with a mixture of metal-labeled antibodies (Supplementary Table 6) for 30 min at room temperature, and washed and incubated with Cell-ID Intercalator-[103]Rh (Fluidigm) overnight at 4 °C. A sample of labeled cells was analyzed using a cytometry by time of flight instrument (Fluidigm) at the MD Anderson Flow Cytometry and Cellular Imaging Core facility.

For flow cytometric analysis, single-cell suspensions were generated from mouse spleens and TdLNs by smashing these tissues with a sterile syringe plunger onto a 100-µm cell strainer; cells were rinsed through the strainer with RPMI/2% fetal bovine serum at room temperature. Regarding the spleens, red blood cells were first removed using RBC Lysis Buffer (Tonbo Bioscience) following the manufacturer's recommendation. Tumors were digested using a mouse Tumor Dissociation Kit. Tumor-infiltrating lymphocytes were enriched on a Ficoll gradient (Sigma-Aldrich). Single-cell suspensions were then incubated with antibodies at $1 \times 10^6$ cells per sample. In all samples, cell surface Fc receptors were blocked via incubation with a rat anti-mouse CD16/32 antibody (1:100; Tonbo Bioscience) for 15 min at 4 °C. Cells were then incubated with fluorescently conjugated antibodies against surface markers for 30 min at 4 °C. For intracellular antibody staining, cells were fixed and permeabilized using commercial reagents (Intracellular Fixation and Permeabilization Buffer Set, eBioscience) and stained with antibodies against intracellular proteins for 30 min at 4 °C. All antibodies were purchased from BioLegend, eBioscience, BD Biosciences, or Tonbo Biosciences (Supplementary Table 9). Dead cells were discriminated in all experiments using Ghost Dye Violet 510 (Tonbo Biosciences). All gating strategy were shown in Supplementary Figs. 8–10. Flow cytometric analysis was performed using LSR II and LSRFortessa flow cytometers (BD Biosciences). Data analysis was performed for viable single cells using FlowJo software (TreeStar).

## Kaplan–Meier plotter analysis

The online Kaplan–Meier plotter database was used to determine the relevance of RNase1 mRNA level to OS of Pan-cancer. Briefly, to create Kaplan–Meier plots of survival curves, Kaplan–Meier plotter database (http://kmplot.com/analysis/) was searched, indicated cancer types (Supplementary Fig. 1f, g) were chosen after clicking the "start KM plotter for pan-cancer", and then gene name was entered. Automatic cutoff scores were selected during queries. Hazard ratio (HRs) and 95% CIs as well as log-rank $P$ values were calculated and displayed on the web page. $P$ values less than 0.05 were considered significant. For the liver hepatocellular carcinoma TCGA data set, ten patients whose pathological diagnoses were reported not HCC (3 cases are fibrolamellar carcinoma; 7 cases are hepatocholangiocarcinoma)[60]. Thus, we described the plot result in Supplementary Fig. 1e as "liver cancer".

## Pull-down assay and immunoblotting

Cells were lysed in RIPA buffer (EMD Millipore) supplemented with Protease and Phosphatase Inhibitor Cocktail (Thermo Fisher Scientific). Lysates were cleared via centrifugation at 13,000 $g$ for 10 min at 4 °C. Supernatants were analyzed for immunoblotting or pull-down assay. For pull-down assay, cell lysates from BMDM or RAW264.6 cells were incubated with or without recombinant mouse RNase1 protein (cat. no. abx068908) at 4 °C for 4 h, and then incubated with GST fusion protein-binding magnetic beads (Thermo Fisher Scientific, #88821) at 4 °C for 8 h. The magnetic beads bound with target proteins were washed with the RIPA buffer and eluted with Blue Loading Buffer (Cell Signaling Technology, # 7722) at 95 °C for 10 min. For immunoblotting, proteins were separated on an 8%, 10%, or 12% Bis-Tris sodium dodecyl sulfate-polyacrylamide gel electrophoresis gel and transferred to polyvinylidene difluoride membranes (Millipore). After blocking with 5% bovine serum albumin, primary antibodies were incubated with the polyvinylidene difluoride membranes in 5% bovine serum albumin overnight at 4 °C. Membranes were washed and hybridized with appropriate secondary antibodies for 1 h at room temperature and imaged using ECL reagents (Bio-Rad Laboratories).

## Immunohistochemistry

Paraffin-embedded HCC patient tissue array slides and HCC tissue slides from animal experiments were stained as described previously[22]. Briefly, HCC samples were incubated with indicated antibodies and then incubated with an avidin-biotin-peroxidase complex. Visualization of the slides was performed using amino-ethyl carbazole chromogen or 3,3′-diaminobenzidine. The immunoreactivity was scored using a well-established immunoreactivity score system in which the score was generated by incorporating both the percentage of positive tumor cells and the intensity of staining[61]. The staining intensity was ranked as negative (score = 0), weak (score = 1), moderate (score = 2), or strong (score = 3). And to ensure the absolute objectivity of the immunohistochemical works in our study, the experiments were conducted in a double-blind manner in which two experienced pathologists stained and evaluated tumor sections independently. Slides in which there was a scoring discrepancy > 10% were reevaluated and reconciled on a two-headed microscope. Final H scores on each case represent the average of scores by each pathologist. Cases with an H-score higher than average were regarded as having high expression, whereas those with an H-score equal to or lower than average were regarded as having low expression.

## Duolink and immunofluorescence assay

A Duolink assay was performed using a Duolink In Situ Red Starter Kit (#DUO92101; Sigma-Aldrich). Briefly, cells were seeded on eight-well chamber slides for 24 h and then treated with RNase1 at a concentration of 1 µg/ml for 15 min or left untreated. Cells were then fixed with 4% paraformaldehyde, permeabilized with 0.5% Triton X-100, and blocked with 1% bovine serum albumin and 22.52 mg/ml glycine in phosphate-buffered saline and 0.1% Tween 20 for 30 min. Cells were then incubated with anti-ALK (cat. no. sc-398791; Santa Cruz Biotechnology) and anti-RNase1 (cat. no. HPA001140; Sigma-Aldrich) antibodies at 4 °C overnight. The following steps were performed according to the Duolink In Situ Red Starter Kit manufacturer's instructions. Distinct red spots were detected using a Zeiss LSM 710 laser microscope, with each spot representing a cluster of protein-protein interactions. For immunofluorescence, liver tumor samples were frozen in an optimal cutting block immediately after extraction. Cryostat sections of samples were mounted on saline-coated slides. Samples were fixed with 4% paraformaldehyde for 15 min at room temperature and blocked with 5% bovine serum albumin, 2% donkey serum, and 0.1 mol/l phosphate-buffered saline at room temperature for 30 min. Sample sections were stained with primary antibodies overnight at 4 °C followed by secondary antibodies at room temperature for 1 h. The secondary antibodies used were anti-rat Alexa Fluor 647 dye conjugate, anti-goat Alexa Fluor 488 dye conjugate, and anti-rabbit Alexa Fluor 594 dye conjugate (Life Technologies). Nuclei were stained with 4,6-diamidino-2-phenylindole (blue; Life Technologies). Sample sections were visualized using a multiphoton confocal laser-scanning microscope (Zeiss LSM 700).

## ELISA

Human and mouse RNase1 plasma concentrations were measured in accordance with the instructions for the Human Ribonuclease A/RNASE1 ELISA Kit (LS-F4615) or mouse RNase1 ELISA kit (LS-F49244). In brief, standard or diluted plasma samples were added to each well and

incubated at 37 °C for 1 h. Next, 100 µl of a biotinylated detection antibody was added to the wells, and the mixtures were incubated at 37 °C for 1 h. After washing three times, 100 µl of a horseradish peroxidase conjugate working solution was added to the wells, and were incubated at 37 °C for 30 min. Following the addition of a 90-µl TMB substrate to the wells and incubation at 37 °C for 15 min, 50 µl of stop solution was added to the wells to terminate coloring, and the optical density of was measured at a wavelength of 450 nm within 20 min. All detection reagents and solutions were provided with the ELISA kit.

### TCGA and RNA-sequencing analysis

RNA-sequencing data obtained using a HiSeq system (Illumina) and the corresponding clinical data for 371 patients with liver cancer were obtained from The Cancer Genome Atlas (TCGA) website (https://portal.gdc.cancer.gov/). Ten patients whose pathological diagnoses were not HCC (3 cases are fibrolamellar carcinoma; 7 cases are Hepatocholangiocarcinoma) and five patients whose survival times were shown as zero were excluded. The Kaplan–Meier plot in Supplementary Fig. 1 was created automatically by KM plotter online database (http://kmplot.com/analysis/). The 370 cases of liver cancer patients were identified from TCGA cohort ($n = 371$) with the available expression level of RNase1, including high ($n = 136$) and low ($n = 234$) levels.

For immune cell infiltrated correlation analysis, normalized RNase1 expression data for HCC cases were obtained from a public TCGA data set, and patients with available follow-up times, gene expression data, and clinical profiles were included. The infiltration levels of immune cell types were quantified by single-sample GSEA (ssGSEA) in R-package gsva. The ssGSEA applies gene signatures expressed by immune cell populations[62] to individual cancer samples[63]. The deconvolution approach used in our study included 24 immunocyte types that could be discriminated according to gene sets specifically overexpressed in each immunocyte. The immunocytes consisted of natural killer (NK) cells, $CD56^{dim}$ NK cells, $CD56^{bright}$ NK cells, plasmacytoid DCs, immature DCs, activated DCs, plasmacytoid DCs, neutrophils, mast cells, eosinophils, macrophages, B cells, T cells, cytotoxic T cells, CD8 + T cells, T helper cells, TCM cells, TEM cells, T follicular helper cells, Tγδ cells, Th1 cells, Th2 cells, Th17 cells, and Tregs. Spearman rank correlation analysis was then performed to measure correlations between the RNase1 expression levels and corresponding immune cell infiltration levels. The ggplot2 package was used to show the plots of immune cell types with correlation coefficients. For M1- and M2-like macrophage infiltration analysis, HCC samples in TCGA were divided into $RNase1^{high}$ ($n = 178$) and $RNase1^{low}$ ($n = 178$) groups according to the average RNase1 expression level.

For RNA-sequencing analysis, next-generation RNA deep sequencing with whole-transcriptome analysis of HCC patient samples ($n = 5$ nivolumab responders, $n = 5$ non-responders) and THP-1 cells under RNase1, LPS/IFN-γ, or IL-4 stimulus was performed using a standard procedure provided by Applied Biosystems. All RNA samples that passed quality tests with RIN values greater than 8 as measured using an Agilent Technologies Bioanalyzer were subjected to RNA deep sequencing. SOLiD fragment colorspace transcriptome reads (50 nt) were mapped to the human genome (hg19) and assigned to ensemble transcripts using Bioscope 1.3.1 (Life Technologies). Differentially expressed genes in different groups were identified using the R-package limma and a cutoff $P$ value of 0.05. Significantly upregulated and downregulated secreted protein-coding genes in nivolumab non-responders are listed in Supplementary Table 2, with an absolute $log_2$ fold change of at least 2 and $P$ value less than 0.001. After TPM transformation, genes with TPM value are 0 in ≥20% samples were excluded to compare TPM value of each sample between non-responders and responders. Principal component analysis was performed using the R-package ggord. For GSEA, the differentially expressed genes in THP-1 cells under treatment with RNase1 or IL-4 were uploaded into GSEA tools and using oncogenic signature gene sets. Positively enriched

pathways with a cutoff false-discovery rate $P$ value less than 0.05 were considered significant pathways. The five most positively enriched pathways in each group were then represented based on their normalized enrichment scores. Gene set variation analysis (GSVA) was performed using "GSVA" package in R software and the reference gene sets were C7: immunologic signatures, which were downloaded from the MSigDB database (https://www.gsea-msigdb.org/gsea/msigdb).

### Cell viability assay

Hep3B and Huh7 Cells were seeded at $1 \times 10^3$ cells/200 ml of medium per well in 96-well plates and treated with or without RNase1 (1 µg/ml) or ALK inhibitors (0.5 µM). After incubation for 48 h, 10 ml of 3-(4,5-dimethylthiazol-2-yl)-2,5-diphenyltetrazolium bromide (5 mg/ml) was added to each well, and the cells were incubated at 37 °C for 3 h. Next, 150 ml of DMSO was added to each well to dissolve the water-insoluble purple precipitate. The absorbance in each well was measured at 595 nm with a reference wavelength of 650 nm using a BioTek Synergy Neo Multi-Mode Microplate Reader.

### Cell invasion assay

BioCoat Growth Factor Reduced Matrigel Invasion Chambers (BD Biosciences) were used to perform a cell invasion assay. Briefly, HCC cells ($1 \times 10^5$ cells/24-well chamber) were cultured with or without indicated treatment in a serum-free culture medium. The chambers were then incubated in a plate with a culture medium containing 10% FBS for 48 h. The cells on the upper surface of the membrane were removed using a cotton swab, and those on the lower surface were fixed with 4% paraformaldehyde, stained with 0.5% crystal violet, and were visualized under a microscope.

### Statistical analyses

Statistical analyses were performed using Prism 8 software (GraphPad Software 8) or the R computing language. Statistical analyses of immune cell numbers, histopathological scores, and human immunohistochemistry scores were performed using a two-tailed Student $t$-test. Body weight changes and tumor size measurements were evaluated using ordinary two-way ANOVA with multiple comparisons tests. Ninety-five percent CIs were calculated.

### Reporting summary

Further information on research design is available in the Nature Portfolio Reporting Summary linked to this article.

## Data availability

RNA-sequencing expression profiles and clinical information for TCGA-HCC patients are publicly available and downloaded from the TCGA database (https://portal.gdc.cancer.gov/). The relevance of gene mRNA level to OS of pan-cancer are publicly available and downloaded from Kaplan–Meier plotter database (http://kmplot.com/analysis/). Immunologic signature gene sets are publicly available and downloaded from the MSigDB database (https://www.gsea-msigdb.org/gsea/msigdb). The immune checkpoint blockade therapy data for gastric cancer (ERP107734) and renal cell carcinoma (RCC) (SRP128156) are publicly available from ICBatlas database (http://bioinfo.life.hust.edu.cn/ICBatlas/#!/). The RNA-seq data for 10 HCC patients and THP-1 cells that used in this study are available in the Gene Expression Omnibus database under accession code GSE215011 and GSE215012. The remaining data are available within the Article, Supplementary Information or Source Data file. Source data are provided with this paper.

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

## Acknowledgements

We thank Dr. Norwood, Donald R who is a scientific editor from the University of Texas MD Anderson Cancer Center for editing the manuscript. This work was funded in part by National Science and Technology Council Taiwan (NSTC 112-2639-B-039-001 -ASP and T-Star Center NSTC 113-2634-F-039-001); Ministry of Health and Welfare Taiwan (MOHW113-TDU-B-222-124016); The Featured Areas Research Center Program by the Ministry of Education (MOE) in Taiwan (to M.-C.H.), the Shanghai International Science and Technology Collaboration Program (18410721900; to N.R.); and the National Natural Science Foundation of China (81472672; to N.R.).

## Author contributions

Conceptualization: C.L., M.H.; Methodology: C.L., C.Z., W.X., Y.Z.; Investigation: C.L., C.Z., W.X., Y.Z., Y.Q., J.W., Q.Z., W.C., Y.W., H.L., S.W., M.K.; Visualization: C.L., C.Z., Y.Z.; Funding acquisition: N.R., M.H.; Project administration: D.Y., M.H.; Supervision: D.Y., N.R., M.H.; Writing – original draft: C.L., M.H.; Writing – review & editing: C.L., M.H.

## Competing interests

The authors declare no competing interests.

## Additional information

[1]Department of Liver Surgery, Center of Hepato-Pancreato-biliary Surgery, Institute of Precision Medicine, The First Affiliated Hospital, Sun Yat-sen University, Guangzhou, Guangdong, China. [2]Department of Molecular and Cellular Oncology, The University of Texas MD Anderson Cancer Center, Houston, TX, USA. [3]Department of Liver Surgery, Liver Cancer Institute, Zhongshan Hospital, and Key Laboratory of Carcinogenesis and Cancer Invasion (Ministry of Education), Fudan University, Shanghai, China. [4]Graduate Institute of Biomedical Sciences, Institute of Biochemistry and Molecular Biology, Research Center for Cancer Biology, Cancer Biology and Precision Therapeutics Center, and Center for Molecular Medicine, China Medical University, Taichung 406, Taiwan. [5]Department of laboratory medicine, Zhujiang Hospital of Southern Medical University, Guangzhou, Guangdong, China. [6]Department of Immunology, The University of Texas MD Anderson Cancer Center, Houston, TX, USA. [7]These authors contributed equally: Chunxiao Liu, Chenhao Zhou. ✉e-mail: liuchx73@mail.sysu.edu.cn; ren.ning@zs-hospital.sh.cn; mhung@cmu.edu.tw

