## [Peer Review File · Nature Communications]

Targeting ALK averts ribonuclease 1-induced immunosuppression and enhances anti-tumor immunity in hepatocellular carcinomaEditorial Note: This manuscript has been previously reviewed at another journal that is not operating a transparent peer review scheme. This document only contains reviewer comments and rebuttal letters for versions considered at *Nature Communications*.

REVIEWERS' COMMENTS

Reviewer #1 (Remarks to the Author):

We appreciate the responses to the prior comments and revisions that the authors have provided. We think that these have significantly strengthened the paper. We have the following comments related to the paper:

Major

105 (Figure 1a-d) We and another reviewer have expressed concern about using this extremely limited dataset as a scientific justification for pursuing RNase1 on a cohort of 10 patients with the previously mentioned confounding issues present in this cohort. If the authors would simply like to explain that they have studied RNase1 and other RNases in the past in the context of other cancers and found the overall survival associations in large cohorts of data intriguing for investigation in combination with this (exceedingly limited) responder/non-responder data, this would be acceptable.

The strong desire to showcase this dataset of PD1 monotherapy-treated patients has significant limitations when there are unquestionably inadequate data. While I do appreciate that a data set of patients treated exclusively with PD-1 monotherapy is unique, this doesn't change the limitations of a 10 patient study. Patients with HCC treated exclusively with immunotherapy are also not rare- Trem/Durva is a standard of care first line regimen by NCCN guidelines. Thousands of patients are receiving this as first line treatment for HCC now. Finally, one could argue that the value of this cohort is further reduced as exceptionally few patients receive PD1 monotherapy- there is not a clinical need to improve a regimen that is not being used in patients. To be clear, we do agree that this data is useful and important, but simply cannot be used as a justification for the pursuit of RNase1. The value it brings does not outweigh its severe limitations.

We acknowledge the IHC analysis of 13 patients that is included from a separate cohort, but again, this seems supplementary rather than foundational in justifying this study.

493 / Discussion: Seeing as the authors have published a paper (PMCID: PMC10321278) demonstrating that RNase1 enhances anti-tumor immunity in a syngeneic model of breast cancer they developed, it would be appropriate to discuss possible reasons for the opposing phenotypes in these models.

Minor

47-48 Sorafenib and Lenvatinib are not the preferred first line recommendations by the NCCN for advanced HCC. In the US, patients do not receive these unless there are contraindications to immunotherapy. Standard of care preferred first line therapies are atezolizumab+bevacizumab or tremelimumab+durvalumab. Second line treatment options include Lenvatinib, sorafenib, cabozantinib, regorafenib, and others.

57-60: This is also outdated. See attached hepatocellular carcinoma clinical guidelines.

100: gene signatures of monocyte to M2-macrophage

116 / Figure 1e/Supp Table 3: I don't see where the numeric overall survival is stated.

130: WRL68 cells

https://www.cellosaurus.org/CVCL_0581

https://www.culturecollections.org.uk/products/celllines/generalcell/detail.jsp?refId=89121403&collection=ecacc_gc

<https://www.cellosaurus.org/str-search/?name=WRL>

68&Amelogenin=X&CSF1PO=9,10&D13S317=12,14&D16S539=9,10&D5S818=11,12&D7S820=8,12&THO1=7&TPOX=8,12&vWA=16,18&

Reviewer #1:

Reviewer #1 (Remarks to the Author):

We appreciate the responses to the prior comments and revisions that the authors have provided. We think that these have significantly strengthened the paper. We have the following comments related to the paper:

Authors' Response: We thank the reviewer for the positive comments. For the additional questions, we have answered as follow:

Major

105 (Figure 1a-d) We and another reviewer have expressed concern about using this extremely limited dataset as a scientific justification for pursuing RNase1 on a cohort of 10 patients with the previously mentioned confounding issues present in this cohort. If the authors would simply like to explain that they have studied RNase1 and other RNases in the past in the context of other cancers and found the overall survival associations in large cohorts of data intriguing for investigation in combination with this (exceedingly limited) responder/non-responder data, this would be acceptable.

The strong desire to showcase this dataset of PD1 monotherapy-treated patients has significant limitations when there are unquestionably inadequate data. While I do appreciate that a data set of patients treated exclusively with PD-1 monotherapy is unique, this doesn't change the limitations of a 10 patient study. Patients with HCC treated exclusively with immunotherapy are also not rare- Trem/Durva is a standard of care first line regimen by NCCN guidelines. Thousands of patients are receiving this as first line treatment for HCC now. Finally, one could argue that the value of this cohort is further reduced as exceptionally few patients receive PD1 monotherapy- there is not a clinical need to improve a regimen that is not being used in patients. To be clear, we do agree that this data is useful and important, but simply cannot be used as a justification for the pursuit of RNase1. The value it brings does not outweigh its severe limitations.

We acknowledge the IHC analysis of 13 patients that is included from a separate cohort, but again, this seems supplementary rather than foundational in justifying this study.

Authors' Response: We thank the reviewer for the comments. We agreed with the reviewer that more patient cases will further ensure the role of RNase1 in immunotherapy. In our story, RNase1 was first identified as a potential factor that was associated with ICI response in two small independent cohorts, and then we used multiple criteria to support the notion, including RNase1-overexpressing and knockout HCC orthotopic mouse model and patient tissue microarray analyses. These pre-clinical studies, tissue microarray data plus the two independent small cohorts pointing out the potential importance for RNase1 as a resistant marker in the future. We understand the cohort size is limited, but that is not the only data present in the story. We will add the following to point out the limitation of small cohort in discussion as following. "We understand the size of cohort is small, but the results are encouraging, and potentially important therefore we performed the following laboratory experiments to provide additional support on the concept."

We also agreed with the reviewer that Durva/Treme (anti-PD-L1 + anti-CTLA-4 combination immunotherapy) has been approved as a standard of care by NCCN guidelines. However, the

roles of CTLA-4 and PD-1 in inhibiting immune responses, including antitumor responses, are largely distinct. Thus, the potential mechanisms associated with monotherapies and combined immunotherapy are quite different. And although few patients receive anti-PD1 monotherapy because the lack of survival benefit, there is clinical need to exploring the resistant mechanism to improve and recommend biomarker-guided combination therapies.

We have added the explanation in the manuscript as follow:

“Considering that RNase1 and other RNases were significantly associated with overall survival in large patient cohorts in different cancer types, RNase 1 was the top candidate for further investigation due to its upregulation in non-responders receiving Nivolumab and its unknown function in tumor immunity.”

493 / Discussion: Seeing as the authors have published a paper (PMCID: PMC10321278) demonstrating that RNase1 enhances anti-tumor immunity in a syngeneic model of breast cancer they developed, it would be appropriate to discuss possible reasons for the opposing phenotypes in these models.

Authors' Response: We thank the reviewer for the suggestion. We have added the discussion in manuscript as follow: “Interestingly, recent results of Wang et al., who found that RNase1 elicits adaptive immune response against breast cancer by boosting CD4⁺ T cell activation through associating with EphA4 (PMCID: PMC10321278). The different binding receptors of RNase1 on macrophage and CD4⁺ T cells might cause the differential responses in the two cell and cancer types. As we know, the number of TAMs in the TME accounts for 20% to 40% among all HCC infiltrating lymphocytes and even more in some rare HCC subtypes (PMID: 36792608). Thus, RNase1 exhibits immunosuppressive effects in our HCC mouse models through activating ALK on TAMs. RNase1 may regulate CD4⁺ T cells activation through binding to different receptor, EPHA4, thus it would be worthwhile to evaluate the proportion of TAMs and CD4⁺ TILs, as well as the expression level of ALK and EphA4 in different patients when these markers are used for choosing appropriate treatment strategy.”

Minor

47-48 Sorafenib and Lenvatinib are not the preferred first line recommendations by the NCCN for advanced HCC. In the US, patients do not receive these unless there are contraindications to immunotherapy. Standard of care preferred first line therapies are atezolizumab+bevacizumab or tremelimumab+durvalumab. Second line treatment options include Lenvatinib, sorafenib, cabozantinib, regorafenib, and others.

Authors' Response: We thank the reviewer for the comments, and we have edited the description as follow: “Multiple strategies have been approved for first or second-line treatment for advanced HCC, such as sorafenib, cabozantinib, lenvatinib, atezolizumab plus bevacizumab, tremelimumab plus durvalumab, and others.”

57-60: This is also outdated. See attached hepatocellular carcinoma clinical guidelines.

Authors' Response: We thank the reviewer for the suggestion. We have modified the description to emphasize the lack of biomarker dilutes the efficacy of current combinational therapies.

“Although several novel combinations are in development, including anti-PD-1/PD-L1 antibodies in combination with anti-CTLA4 and/or anti-VEGF antibodies or multikinase inhibitors, the lack of predictive biomarkers to guide the use of those combinational strategies dilute the efficacy.”

100: gene signatures of monocyte to M2-macrophage

Authors' Response: Transcriptional data (deposited in GEO, GSE5099) revealed differences in monocyte to M2-macrophage gene signatures. The entire data sets are available at the Gene Expression Omnibus (GEO) website (www.ncbi.nlm.nih.gov/geo), accession number GSE5099. The accession numbers of each pathway have been shown in supplemental figure 1c.

116 / Figure 1e/Supp Table 3: I don't see where the numeric overall survival is stated.

Authors' Response: We have added the value of mean overall survival (months) in Supplemental Table 3. The mean level of OS in RNase1^{low} vs RNase1^{high} patients is 51.16±1.87 vs 40.07±2.28.

130: WRL68 cells

Authors' Response: We thank the reviewer for the update information of WRL68 cells. We have removed the description of WRL68 cells in text and only showed the expression of RNase1 in HCC cell lines in Fig 1f.